# Glial immune-related pathways mediate effects of closed head traumatic brain injury on behavior and lethality in *Drosophila*

**Bart van Alphen**[1,☯,†]**, Samuel Stewart**[1,☯,¤a]**, Marta Iwanaszko**[1,2]**, Fangke Xu**[1]**, Keyin Li**[1]**, Sydney Rozenfeld**[1]**, Anujaianthi Ramakrishnan**[1]**, Taichi Q. Itoh**[1,¤b]**, Shiju Sisobhan**[1]**, Zuoheng Qin**[1]**, Bridget C. Lear**[1]**, Ravi Allada**[1] *

**1** Department of Neurobiology, Northwestern University, Evanston, Illinois, United States of America,
**2** Department of Preventive Medicine—Biostatistics, Feinberg School of Medicine, Northwestern University, Chicago, Illinois, United States of America

☯ These authors contributed equally to this work.
† Deceased.
¤a Current address: Department of Neuroscience, Baylor College of Medicine, Houston, Texas, United States of America
¤b Current address: Faculty of Arts and Science, Kyushu University, Nishi-ku, Fukuoka, Japan
* r-allada@northwestern.edu

**Academic Editor:** The PLOS Biology Editors

**Data Availability Statement:** The code used to generate the results that are reported in this study are available from allada-lab@northwestern.edu upon reasonable request. Data supporting the findings are available in the respective

## Abstract

In traumatic brain injury (TBI), the initial injury phase is followed by a secondary phase that contributes to neurodegeneration, yet the mechanisms leading to neuropathology in vivo remain to be elucidated. To address this question, we developed a *Drosophila* head-specific model for TBI termed *Drosophila* Closed Head Injury (dCHI), where well-controlled, nonpenetrating strikes are delivered to the head of unanesthetized flies. This assay recapitulates many TBI phenotypes, including increased mortality, impaired motor control, fragmented sleep, and increased neuronal cell death. TBI results in significant changes in the transcriptome, including up-regulation of genes encoding antimicrobial peptides (AMPs). To test the in vivo functional role of these changes, we examined TBI-dependent behavior and lethality in mutants of the master immune regulator NF-κB, important for AMP induction, and found that while sleep and motor function effects were reduced, lethality effects were enhanced. Similarly, loss of most AMP classes also renders flies susceptible to lethal TBI effects. These studies validate a new *Drosophila* TBI model and identify immune pathways as in vivo mediators of TBI effects.

**Editor's Note**

During the second round of review of this article, it came to light that the first author, under a pseudonym, had engaged in activities which do not align with PLOS' values. Please note that PLOS strongly condemns all discriminatory behaviors, attitudes, and actions.

supplemental data files, while the NGS data files were deposited in the Gene Expression Omnibus (GEO) under the accession code GSE164377.

**Funding:** This study was funded by the Department of Defense (W81XWH-20-1-0211 and W81XWH-16-1-0166 to RA) and the Defense Advanced Research Projects Agency (D12AP00023). The funders had no role in study design, data collection and analysis, decision to publish, or preparation of the manuscript.

**Competing interests:** The authors have declared that no competing interests exist.

**Abbreviations:** alrm, *astrocytic leucine-rich repeat molecule*; ALS, amyotrophic lateral sclerosis; AMP, antimicrobial peptide; DAM, *Drosophila* activity monitoring; DAMP, damage-associated molecular pattern; DAT, dopamine transporter; dCHI, *Drosophila* Closed Head Injury; DE, differential expression; elav, embryonic lethal abnormal visual system; GO, gene ontology; Imd, Immunodeficiency; IP, immunoprecipitate; JAK–STAT, Janus Kinase protein and the Signal Transducer and Activator of Transcription; MMP-1, matrix metalloproteinase-1; NF-κB, nuclear factor kappa B; nsyb, synaptobrevin; Rel, *Relish*; repo, *reversed polarity*; TBI, traumatic brain injury; Tot, Turandot; TRAP-Seq, Translating Ribosome Affinity Purification and Sequencing; UQ, upper-quartile; vir-1, virus-induced RNA-1.

After becoming aware of this issue, we took into consideration that the other authors of the article may have been unaware of the first author's activities, that the research reported in this article was likely unaffected by them, and that the issue had come to light after the authors had invested significant effort and resources in conducting this study and revising the manuscript to address issues raised by *PLOS Biology* reviewers. Given these factors, we decided to continue considering the article for publication.

*PLOS Biology* has tried to ensure that this article's evaluation was not affected by competing interests and adhered to the journal's high standards for fair, rigorous, and objective peer review. Given the details of this case we have honored the positions of editors when they requested anonymity. The *PLOS Biology* Staff Editors are listed as handling editors on this article for this reason. The peer review process involved the *PLOS Biology* Staff Editors and six external subject matter experts, including two members of our Editorial Board.

### Authors' Note

The research presented here was conducted over several years by a large diverse and multidisciplinary group of scientists. We (the co-authors) became aware of the offensive online posts made by Dr. Van Alphen only shortly before his death, while the manuscript was already in revision. We were shocked by these comments and condemn them as anathema to our core values.

We have chosen to publish the science that includes Dr. Van Alphen's work, as well as significant contributions of several authors, since it addresses an important scientific and societal problem, and since it was conducted with grant funding that carries the responsibility to communicate scientific discoveries with the broader community. The science presented here is independent of the personal views of any of the investigators.

## Introduction

Traumatic brain injury (TBI) is one of the major causes of death and disability in the developed world [1–3]. Yet, the underlying mechanisms that lead to long-term physical, emotional, and cognitive impairment remain unclear.

Unlike in most forms of trauma, a large percentage of people killed by TBIs do not die immediately but rather days or weeks after the insult [4]. The primary brain injury is the result of an external mechanical force, resulting in damaged blood vessels, axonal shearing [5], cell death, disruption of the blood–brain barrier, edema, and the release of damage-associated molecular patterns (DAMPs) and excitotoxic agents [6]. In response, local glia and infiltrating immune cells up-regulate cytokines (tumor necrosis factor α) and interleukins (IL-6 and IL-1β) that drive posttraumatic neuroinflammation [7–10]. This secondary injury develops over a much longer time course, ranging from hours to months after the initial injury and is the result of a complex cascade of metabolic, cellular, and molecular processes [11–13]. Neuroinflammation is beneficial when it is promoting clearance of debris and regeneration [14] but can become harmful, mediating neuronal death, progressive neurodegeneration, and neurodegenerative disorders [15–18]. The mechanisms underlying these opposing outcomes are largely unknown but are thought to depend of the location and timing of the neuroinflammatory response [19,20]. It remains to be determined what the relative roles of TBI-induced neuroinflammation and other TBI-induced changes are in mediating short- and long-term impairments in brain function in vivo.

To study the mechanisms that mediate TBI pathology in vivo over time, we employ the fruit fly *Drosophila melanogaster*, a model organism well suited to understanding the in vivo genetics of brain injury. Despite considerable morphological differences between flies and

mammals, the fly brain operates on similar principles through a highly conserved repertoire of neuronal signaling proteins, including a large number of neuronal cell adhesion receptors, synapse-organizing proteins, ion channels and neurotransmitter receptors, and synaptic vesicle-trafficking proteins [21]. This homology makes *Drosophila* a fruitful model to study neurodegenerative disorders [22], including amyotrophic lateral sclerosis (ALS) [23], Alzheimer disease [24], Huntington disease [25], and Parkinson disease [26].

Trauma-induced changes in glial gene expression are a highly conserved feature of both mammalian [27,28] and *Drosophila* glia [29–32] (reviewed in [33]). In *Drosophila*, glia are able to perform immune-related functions [32,34]. Ensheathing glia can act as phagocytes and contribute to the clearance of degenerating axons from the fly brain [29,31,35]. The *Drosophila* innate immune system is highly conserved with that of mammals and consists primarily of the Toll, Immunodeficiency (Imd), and Janus Kinase protein and the Signal Transducer and Activator of Transcription (JAK–STAT) pathways, which, together, combat fungal and bacterial infections [36,37]. Dysregulation of cerebral innate immune signaling in *Drosophila* glial cells can lead to neuronal dysfunction and degeneration [38,39], suggesting that changes in glia cells could underlie secondary injury mechanisms in our *Drosophila* model of TBI.

Most *Drosophila* TBI models [40,41] deliver impacts to the entire body, not just the head, and thus, one cannot definitively attribute ensuing phenotypes to TBI. More recently, a *Drosophila* TBI assay was published that uses head compression in flies just recovered from anesthesia to induce TBI [42]. To remove the confounds of bodily injury and anesthesia, we have developed a head-specific *Drosophila* model for TBI, *Drosophila* Closed Head Injury (dCHI). Here, we show that by delivering precisely controlled, nonpenetrating strikes to an unanesthetized fly's head, we can induce cell death and increased mortality in a dose-dependent manner. In addition, TBI results in impaired motor control and decreased, fragmented sleep in flies that survive the injury. Impaired motor control persists for many days after TBI, while the sleep phenotype disappears after 3 days. In wild-type flies, TBI results in changes in glial gene expression, where many immune-related genes, including most antimicrobial peptides (AMPs) are up-regulated 24 to 72 hours after injury. TBI-induced behavioral phenotypes do not occur in mutants lacking the master immune regulator nuclear factor kappa B (NF-κB) *Relish* (*Rel*), even though TBI-induced mortality is greatly induced in these mutants, suggesting that these impairments are due to immune activation rather than the injury itself. CRISPR deletions of most AMP classes increase TBI-induced mortality, but survival is increased in flies lacking Defensin, suggesting that the innate immune response to TBI in *Drosophila* can have both beneficial and detrimental effects. Together, these results establish a platform where powerful *Drosophila* genetics can be utilized to study the complex cascade of secondary injury mechanisms that occur after TBI in order to genetically disentangle its beneficial and detrimental effects.

## Methods

### Flies

Fly stocks were raised on standard cornmeal food under a 12-hour light/12-hour dark cycle at 25°C and approximately 65% relative humidity. TBI inductions and climbing assays were carried out in the lab at room temperatures (approximately 21 to 23°C). For sleep and life span experiments, flies were kept on standard cornmeal food under a 12-hour light/12-hour dark cycle at 25°C and approximately 65% relative humidity. All experiments were carried out in young adult (3 to 7 days old) male iso31 flies, an isogenic $w^{1118}$ control strain commonly used for sleep research. NF-κB Relish null mutants (Relish[E20]) were obtained from Bloomington ($w^{1118}$; Rel[E20] e[s]; #9457). Repo-Gal4 was obtained from Bloomington (w[1118]; P{w[+m*] = GAL4repo/TM3, Sb [1] #7415). *UAS-GFP::RpL10A* was obtained from the Jackson lab [43].

CRISPR deletions of different classes of AMPs were obtained from Bruno LeMaitre and compared to their iso31 control strain {Hanson, 2019}. For the glial RNAi screen, RNAi lines obtained from Bloomington and VDRC were crossed to repo-Gal4 (BDRC# 7415). Controls consist of the appropriate RNAi control line (y [1] v [1]; P{y[+t7.7] = CaryP}attP2 (BDRC# 36303) for attP2 TRiP lines; y [1] v [1]; P{y[+t7.7] = CaryP}attP40 (BDRC# 36304 for attP40 TRiP lines; isogenic host strain w1118, for GD lines (VDRC ID 60000), empty insertion line y, w[1118];P{attP,y[+],w[3']}} (VDRC ID 60100) for KK lines). Attacin-A (BDRC 56904, VDRC 50320GD), Attacin-B (VDRC 57392, VDRC 33194GD), Attacin-C (VDRC 101213KK, VDRC 42860GD), Cecropin-A1 (BDRC 64855), Cecropin-A2, (BDRC 65160), Cecropin-B (BDRC 61932), Cecropin-C (BDRC 50602), Diptericin-A (BDRC 53923, VDRC 41285GD), Diptericin-B (BDRC 28975), Drosocin (BDRC 67223, VDRC 42503), Drosomycin (BDRC 55391, 63631), Listericin (VDRC 102769 KK), Metchnikowin (BDRC 28546, VDRC 109740 KK), virus-induced RNA-1 (vir-1) (BDRC 58209). All flies were collected under $CO_2$ anesthesia at least 24 hours before TBI induction and placed on regular food.

## Aspirator and fly restraint assembly

Aspirators were constructed by wrapping a small square of cheesecloth around one end of aquarium tubing. A P1000 pipette tip was securely attached to covered end of the tubing, and the tip of the pipette tip was cut off to leave an aperture large enough for an individual fly to pass through without difficulty. The aspirator is used to transport individual flies via mouth pipetting. This allows flies to be transferred from their home vials to the experimental setup without using anesthesia. Fly restraints were created by cutting off the last 3 to 4 millimeters of P200 pipette tips to create an aperture large enough to let an individual fly's head through without letting the entire body through. Multiple sizes of fly restraints were produced to accommodate small variations in size among flies.

## *Drosophila* closed head TBI assay

Flies were removed from their home vials without the use of anesthetic, using an aspirator and gently transferred to a prepared P200 pipette (see above). By applying some air pressure on the aspirator, the fly is pushed into the P200 pipette in such a way that the fly gets stuck at the end, with only its head sticking out. The restrained fly is then placed in a micromanipulator allowing for movement in 3 dimensions, which was subsequently used to move the fly into the appropriate position, with the back of the fly's head making contact with the pin of a pull-type solenoid (uxcell DC 12V), which delivers 8.34 Newtons of force. Flies were observed using a high-powered camera lens (Navitar Zoom 6000, Rochester, NY) to ensure that they were in the proper position. A variable-voltage power supply (Tenma Corporation, Tokyo, Japan) was set to 12 V and used to power the solenoid, which then delivered a blow to the fly's head (Fig 1). Flies were hit 1 time, 5 times, and 10 times when observing effect of number of blows on response to TBI. Flies were hit 5 times for all other experiments. To minimize confounding effects of anesthesia, flies were collected under $CO_2$ anesthesia at least 24 hours before each experiment. All experiments are carried out in awake, unanesthetized flies.

## Quantifying locomotor behavior immediately after TBI

Immediately after TBI induction, individual flies were placed in 35 mm petri dishes, along with some fly food. Fly positions were recorded at 5 frames per second for 4 consecutive hours using a Blackfly CCD camera (FLIR Systems, Wilsonville, OR). Video data were analyzed using a custom Matlab script, using background subtraction to find fly positions in each frame, from which we derived velocity, latency to move, and the percentage of time each fly is active.

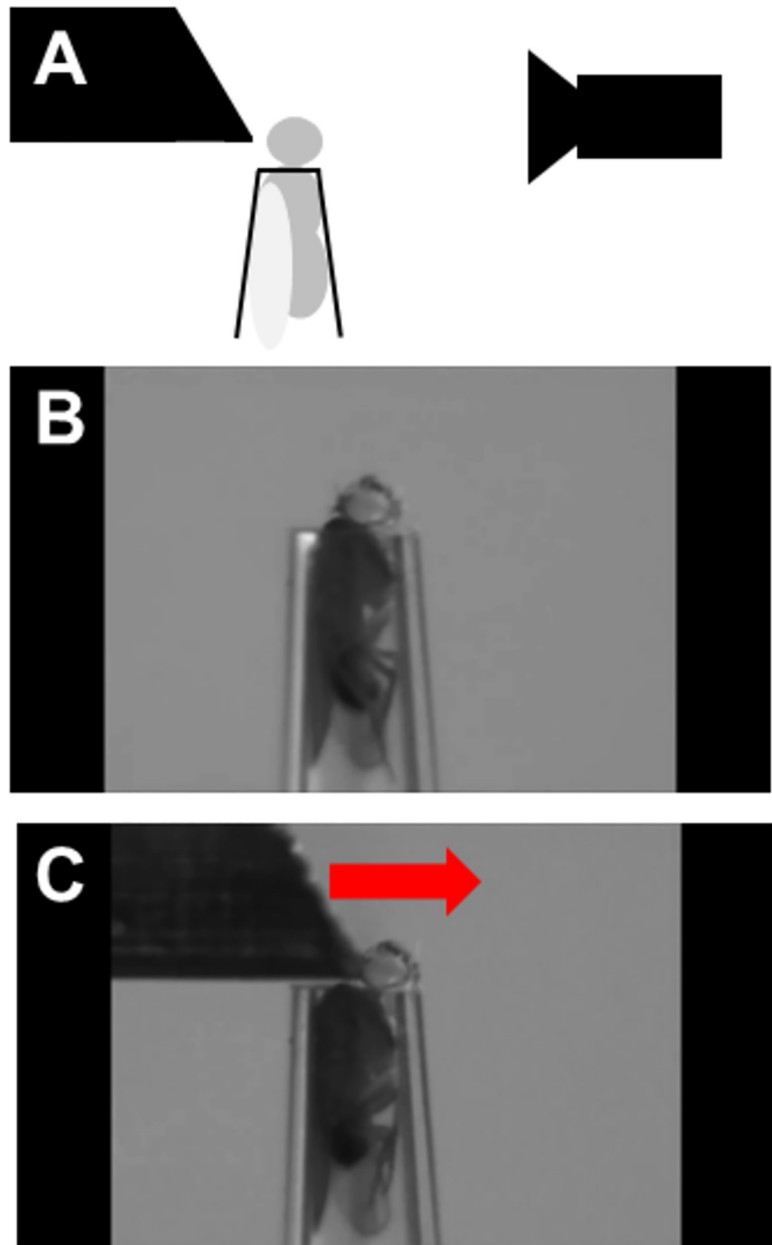

**Fig 1. Single fly TBI setup. (A, B)** To induce head-specific TBI, individual flies are gently aspirated into a modified 200-μL pipette tip that acts as a restraint. Immobilized flies are placed in front of a solenoid, using a set of micromanipulators with 5 degrees of freedom (x,y,z, pitch, roll) and a high-magnification video system to ensure highly replicable positioning. TBI is induced by running a current through the magnetic coil of the solenoid, which retracts a brass trapezoid-shaped block. **(C)** By releasing current, a spring drives the brass block forward, hitting the fly on the top of the head. TBI, traumatic brain injury.

### Negative geotaxis response

A climbing assay is used to measure locomotor deficits after TBI in a manner similar to the RING assay [44]. Flies were individually stored in food vials and kept under the conditions discussed above. Vials were vertically divided into six 1-cm tall segments, labeled in order of ascending height (0 cm, 1 cm, etc.). Vials were tapped on a lab bench as a startle stimulus. Flies

were then allowed to climb freely for 4 seconds, after which the highest point reached by the flies was observed and recorded. Three trials were observed for each individual fly; flies were allowed a period of at least 1 minute of recovery in between trials. Measurements from individual flies' trials were then averaged to calculate a fly's mean performance.

## Mortality assay

After TBI induction, flies are housed in plastic vials with standard corn meal medium and housed in a 12-hour light/12-hour dark cycle at 25˚C and approximately 65% relative humidity. Flies are gently transferred to fresh vials every 3 days. Deceased flies remaining in the old vial are counted.

## Sleep assay

Three- to 7-day-old flies were placed into individual 65-mm glass tubes in the *Drosophila* activity monitoring (DAM) system (Trikinetics, Waltham, MA), which were placed in incubators running a 12-hour light/12-hour dark cycle. All experiments were carried out at 25˚C. Sleep data were collected by the DAM system in 1-minute bins and analyzed offline using custom-made Matlab scripts (Matlab 2011a, Mathworks, Natick, MA). Briefly, sleep was defined as any period of inactivity of 5 minutes or more [45,46]. For each fly, total amount of sleep per day, average bout length, number of sleep bouts, number of brief awakenings, and average daily activity were derived from its activity trace (number of infrared beam crossings per minute).

## Statistics

All statistical analysis for behavioral experiments was performed using Matlab 2011a for PC. For TRAP-seq analysis, see below. Mortality assay: Survival curves were plotted using the Kaplan–Meier estimator as described [47]. The statistical significance was calculated using the log-rank test. Plots and log-rank tests were performed in Matlab, using scripts developed by [48].

## TUNEL assay

A TUNEL assay was performed in whole brain as per manufacturer's protocol (In situ cell death detection kit, Fluorescein, Sigma Aldrich, St. Louis, MO). The brains were carefully dissected out at different time points and fixed in 4% paraformaldehyde for 20 minutes followed by 3× 15-minute wash in PBST (PBS with 0.5% Triton-X 100). The brains were incubated in TUNEL mixture (prepared as per manufacturer's instruction) for 60 minutes at 37˚C followed by 3× 15-minute wash in PBST. The brains were then mounted in Vectashield mounting medium. TUNEL-positive values were determined for the entire central brain.

## TRAP-seq

After receiving TBI, flies were collected at one of 3 time points, namely, 1 day postinjury, 3 days postinjury, and 7 days postinjury at ZT0 (lights-on in 12-hour light:12-hour dark). Flies were collected in 15 ml conical tubes and flash frozen in liquid nitrogen. Their heads were collected by vigorously shaking frozen flies and passing them through geological sieves. Approximately 100 heads were used for each experiment. Heads were homogenized for 3 minutes by Pellet Pestle Cordless Motor. Translating Ribosome Affinity Purification and Sequencing (TRAP-Seq) was performed as described [43,49]. Sepharose beads were prepared by rinsing 25 μL of resin per reaction with 1 mL of extraction buffer. Protein A Plus UltraLink (PAS)

resin was incubated with 1 mL of extraction buffer and 2.5 uG of HTZ 19C8 antibody and rotated for 2 to 3 hours at room temperature. Beads were then spun at 2,500$g$ for 30 seconds at room temperature and rinsed another 3 times with extraction buffer. The conjugated beads were then incubated with 1 mL of blocking buffer for 15 minutes at 4˚C. The beads were then spun again at 2,500$g$ for 30 seconds, and the supernatant was discarded. The beads were washed with 1 ml cold extraction buffer. This process was repeated another 2 times. Beads were incubated with 260 μL of head extract for 1 hour at 4˚C and then spun at 2,500$g$ for 30 seconds at 4˚C. The beads were rinsed with 1 mL of cold wash buffer at 4˚C. This process was repeated 3 times. After the final wash, 1 mL of Trizol was added. The beads were rotated at room temperature for 15 minutes. Chloroform was added, and the beads were subsequently shaken by hand for 30 seconds and incubated for 3 minutes at room temperature. The beads were then centrifuged at 15,000 rpm for 15 minutes at 4˚C. The resulting upper aqueous phase was extracted and transferred to a new tube with 70% ethanol. RNA was extracted following the RNeasy Micro Kit protocol (Qiagen, Venlo, the Netherlands). RNA purified from the GFP tagged RpL10 was then reverse transcribed to cDNA. The cDNA was used as template for T7 transcriptase to amplify the original RNA. We synthesized first and second strand cDNA from RNA first with Superscript III and DNA polymerase. Then, we amplified the RNA by synthesizing more RNA from the cDNA template with T7 RNA polymerase. Amplified RNA was purified with RNeasy Mini Kit (Qiagen). A detailed procedure for amplification can be in found in [50]. After the second round of cDNA synthesis from amplified RNA, the cDNA was submitted to HGAC at University of Chicago for library preparation and sequencing.

## Quantification of data, differential expression, and functional annotation analyses

Sequencing was done with Illumina HiSeq 2000. All samples are done with single-end reads of 50 base pairs in length. At least approximately 5,000,000 mappable reads were obtained and used for quantification for each sample. Reads were quantified against transcript assembly release 6.10 from Flybase with Kallisto. Results of each gene were calculated by adding up all the transcripts for the gene. RNA-seq data were quantified at transcript level using Kallisto [51], using FlyBase_r6.14 as a reference transcriptome [52]. Quantified transcripts were summed up to the gene level using tximport library [53]. A minimal prefiltering, keeping only rows with more than 2 reads, was applied to gene level data before differential expression (DE) analysis. Differential gene expression analysis was performed on TBI versus control data with DESeq2 [54], using the likelihood ratio test to correct for batch effect among the biological replicates. Genes with the absolute log2 fold change higher than 0.6, and false discovery rate adjusted $p$-values ≤ 0.1 were identified as differentially expressed consistent with previous studies [55–60]. Day 7 replicates were corrected for sequencing depth and possibly other distributional differences between lanes, using upper-quartile (UQ) normalization, available through RUVSeq library [61], before proceeding to DE analysis. One replicate was removed from further analysis, due to extremely low expression across the sample, which was not comparable to the levels observed in the other Day 7 replicates. DEseq2 can assess DE with 2 replicate samples as input [62–67]. Functional annotation of DE genes was performed using the DAVID database (release 6.8 [68,69]) with a focus on gene ontology (GO) terms and Reactome pathways. Fastq data have been uploaded to the GEO repository (Series record GSE164377).

Supporting figures for post-TBI days 1, 3, and 7 (S5 Fig) show sample comparison of relative log expression in untreated and successfully corrected data (panels A, C, E and B, D, F, respectively). Small deviations, arising from the technical differences, can be observed in D01 and D03; these were removed with UQ between lane correction [70]. For D07 (S5E Fig), we

have observed that the replicates are lower quality, and there is a significant deviation in values between replicates within the TBI group, with replicate R3 assumed to be corrupted (see S5 Fig). For consistency, we applied the same correction method to remove technical differences from post-TBI day 7, but as expected, replicate R3 did not improve. Taking this into consideration, we decided to remove this replicate from further analysis.

## Results

### dCHI: A controlled head impact model for TBI in *Drosophila*

To study TBI in flies, we developed a head-specific TBI model where brain injury is inflicted in unanesthetized, individually restrained flies using a solenoid to deliver well-controlled, nonpenetrating strikes to the fly head (Fig 1). For TBI induction, individual flies are transferred from their home vial to a prepared P200 pipette tip, using an aspirator. Flies are gently blown upward until the head emerges from the tip of the pipette (Fig 1B). The pipette is then placed in a micromanipulator platform with 5 degrees of freedom (pitch, roll as well as movement along the XYZ axes). The top of the fly head is pressed against the tip of the solenoid that consists of a metal pin running through a copper coil attached to a spring. By running a current through the coil, it acts as a magnet, drawing the pin back and arming the spring. When the current is halted, the spring causes the pin to shoot out, thus allowing us to deliver one or more blows to the fly's head (Fig 1C, S1 Movie). After TBI induction, flies are aspirated out of the pipette tip and returned to an empty vial containing regular fly food.

### dCHI results in immediate locomotor defects

Immediately after TBI induction, flies are often able to stand but only barely respond to tactile stimuli (S2 Movie). However, mobility returns in a manner of minutes (S3 Movie).

To quantify locomotor impairments after TBI, we placed flies in 35 mm petri dishes immediately after TBI, along with some fly food, and recorded fly positions using a CCD camera. Sample traces for 3 intensities (1, 5, and 10 strikes; TBIx1, TBIx5, and TBIx10) as well as sham-treated controls are shown in S1A Fig.

After TBI, approximately 25% flies in the TBIx1 condition are immobile versus approximately 55% in the TBIx5 and TBIx10 conditions (S1B Fig). Flies in the TBIx1 condition started moving within seconds, while flies in the TBIx5 and TBIx10 conditions started moving after minutes (3.3 and 10 minutes, respectively; S2C Fig). We also observed some locomotor defects (circling, slow walking, sideways walking, backwards walking, and jumping) shortly after TBI onset, in a dose-dependent manner (25%, 45%, and 50% in the TBIx1, TBIx5, and TBIx10 groups, respectively) (S2D Fig). These movement disorders only occurred in flies that were immobile immediately after TBI and were not observed in flies that immediately started walking. Walking speed was reduced in all 3 groups during the first hour post-TBI, but the TBIx1 and TBIx5 groups had recovered by the second hour. Walking speed remained impaired for all 4 hours in the TBIx10 group (S2E Fig). Overall activity (% of time active) was significantly reduced in the TBIx5 and TBIx10 groups for the first hour after TBI but unaffected in the TBIx1 group (S2F Fig).

### dCHI increases mortality and impairs negative geotaxis in a dose-dependent manner within 24 hours

We next examined the pathological and behavioral effects within the first 24 hours post-dCHI. TBI phenotypes become more severe with consecutive strikes in mammals [71] and *Drosophila* [40,41]. We subjected male flies to 1, 5, or 10 consecutive solenoid strikes, delivered at 1 strike

per second. After TBI induction, treated and sham-treated cohorts were individually housed in vials containing standard food. Twenty-four hours after TBI exposure, surviving flies were counted in each of the 4 groups. We observed a dose-dependent increase in 24-hour mortality (Fig 2A). At 1 strike (TBIx1), there is no effect on 24-hour mortality ($p = 0.68$). Mortality is increased in a dose-dependent manner (control versus TBIx5, $p = 0.03$; control versus TBIx10, $p = 0.004$; ANOVA with Dunnett post hoc test, F(3,8) = 8.41; $n = 3$ replicates of 10 flies/ group).

Loss of balance and poor motor coordination are symptoms associated with TBI [72–74]. Impairments in motor control, balance, and sensorimotor integration are also a well-studied endophenotype in rodent models of TBI (as quantified by beam balance, beam walk, and rotarod assays; reviewed in [75]). In *Drosophila*, impairments in sensorimotor integration are quantified by measuring the negative geotaxis response, a reflexive behavior where a fly moves away from gravity's pull when agitated [76]. Impaired negative geotaxis has been observed in aging and in *Drosophila* models of neurodegeneration [77–79].

To assess sensorimotor function after TBI, we used a variation of the negative geotaxis assay [44], where the average height climbed in a defined time period is quantified, rather than a pass/fail number for absolute height as more subtle deficits can be observed using this approach. Typically, young adult wild-type flies reach an average climbing height of approximately 4 to 5 cm in a 3-second time period [44]. In our assay, sham-treated $w^{1118}$ flies (3- to 7-day-old males) reached an average height of approximately 3.4 cm in 4 seconds (Fig 2B; 0 days post-TBI). Climbing behavior, driven by negative geotaxis becomes impaired after TBI. After a single hit, there is no detectable difference in climbing, 24 hours after TBI induction (control versus TBIx1, $p = 0.1876$; Fig 2B). However, after 5 or 10 consecutive hits, climbing behavior becomes impaired in a dose-dependent manner (Fig 2B; control versus TBIx5, $p = 2.67 \times 10^{-6}$; control versus TBIx10, $p = 2.67 \times 10^{-6}$; ANOVA with Dunnett post hoc test, F (3,99) = 57.54; $n = 30$ flies/group).

## TBI increases apoptotic cell death in a dose- and time-dependent manner

To test whether our TBI assay causes neuronal death, apoptosis was quantified using a TUNEL assay [80] after inducing TBI by striking flies either 5 or 10 times and comparing the number of TUNEL-positive cells at 3 different time points (4, 8, and 24 hours) between TBI-treated flies and sham-treated controls. Controls showed, on average, 2 to 4 TUNEL-positive cells, which may be spontaneous apoptotic cells (Fig 2C and 2D). Four hours after TBI induction, we saw an increase in TUNEL-positive cells in the TBIx10 condition ($p = 2.56 \times 10^{-6}$) but not in the TBIx5 condition ($p = 0.1027$; F(2,23) = 68.29) at this time point (Fig 2D). Eight hours after TBI induction, we also saw an increase in TUNEL-positive cells in the TBIx10 condition ($p = 2.93 \times 10^{-6}$) but not in the TBIx5 condition ($p = 0.5623$; F(2,22) = 33.41) at this time point (Fig 2D). Twenty-four hours after TBI induction, we saw an increase in TUNEL-positive cells in both the TBIx5 ($p = 2.57 \times 10^{-6}$) and the TBIx10 condition ($p = 2.53 \times 10^{-6}$; F(2,19) = 111.23) at this time point (Fig 2D). ANOVA with Dunnett post hoc test. Taken together, dCHI induces advanced mortality, motor deficits, and cell death within the first 24 hours.

## dCHI reduces life span

Given the slowly evolving nature of TBI pathology, we next examined the chronic effects of dCHI over time. We first examined life span. Unlike other forms of trauma, death after TBI rarely occurs immediately. To test how our TBI assay affects overall life span, we delivered 5 consecutive strikes to the top of a fly's head (S1 Movie). After this, flies were housed individually, and survivors were counted every day. dCHI significantly reduces life span (log-rank test

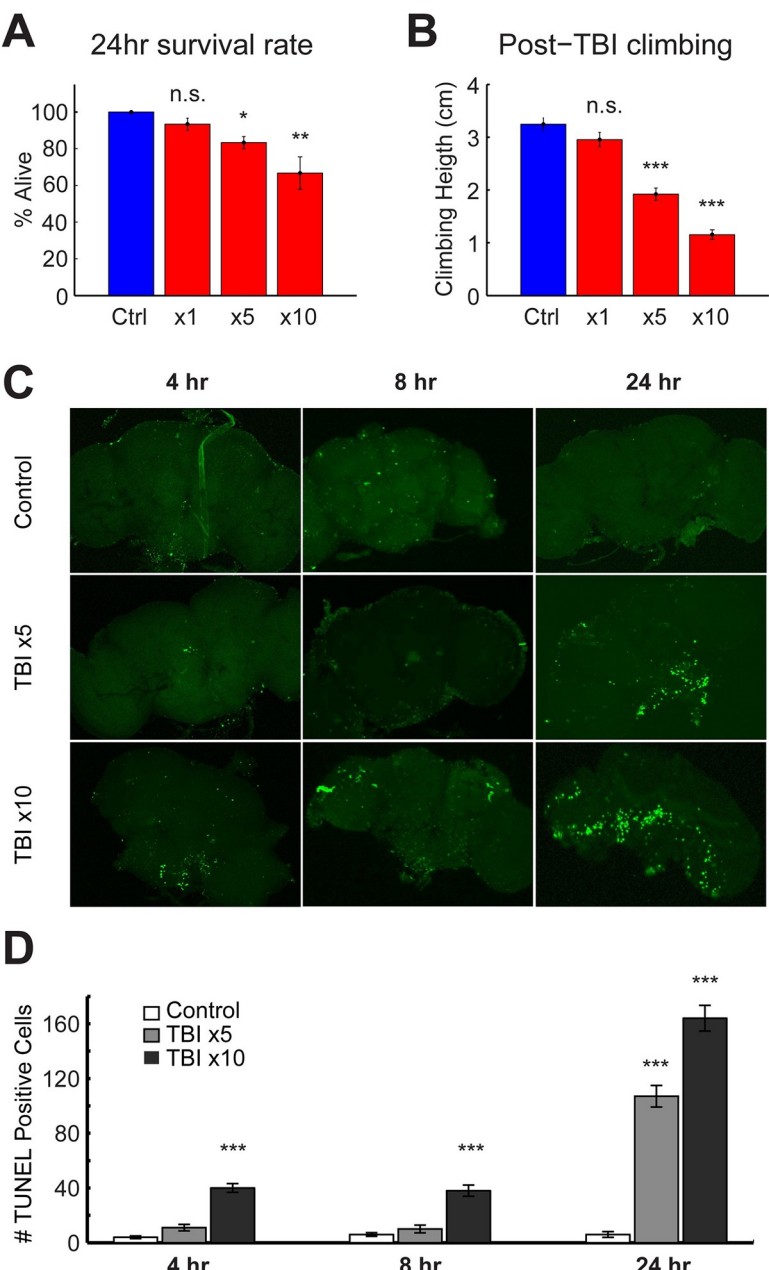

**Fig 2. TBI causes cell death, mortality, and impaired climbing in a dose-dependent manner.** Male w1118 flies were exposed to either 1, 5, or 10 strikes to the head, delivered at 1 strike per second (*n* = 32 per group). (**A**) 24-hour survival rate decreased with increased number of strikes. (**B**) In surviving flies, climbing behavior was quantified and compared to sham-treated controls 24 hours after TBI. Climbing behavior became more impaired with increased TBI severity (n. s = not significant, *** $p < 0.001$, one way ANOVA with Dunnett post hoc test, *n* = 30/group). Cell death following TBI was quantified with a TUNEL assay. (**C**) Representative images of TUNEL staining at different time points in control and post-TBI flies. (**D**) Histogram showing significantly increased TUNEL-positive cells post-TBI in a dose-dependent manner (*n* = 10, 8, 8 for controls, *n* = 8, 8, 9 for TBIx5, and *n* = 7, 7, 8 for TBIx10 at 4 hours, 8 hours, and 24 hours, respectively). * $p < 0.05$, ** $p < 0.01$, *** $p < 0.001$ ANOVA with Dunnett post hoc test. Error bars indicate SEM. All figure-related data are located in S2 Data. TBI, traumatic brain injury.

on Kaplan–Meier survival curves, $p < 0.001$). Around 50% of the TBI group had died 14 days after TBI induction, while 50% of the sham-treated controls had died 32 days after the start of the survival assay (Fig 3A).

To test whether this increase in mortality is mainly due to flies dying during the first 14 days after TBI, we removed flies that died during this peropd from both controls and TBI flies cumulatively, for up to 2 weeks after TBI, and performed log-rank test on the remaining flies. In all cases, survival rate is still significantly decreased in the TBI group, suggesting that the increased mortality is not due to flies that die early (S2 Fig).

## dCHI impairs motor control in a biphasic manner

One day after being subjected to 5 consecutive hits, injured flies display a decrease in climbing capacity, compared to sham-treated controls (Fig 3B; $p < 0.001$, $t$ test with Bonferroni correction). Flies recover from days 2 to 3, suggesting that climbing deficits are not due to permanent injuries to the legs or to Johnston's organ, the fly's gravity sensor [81,82]. Subsequently, flies undergo a relapse as climbing behavior is impaired again on days 4 to 7 ($p < 0.01$, $t$ tests with Bonferroni correction). The biphasic response to dCHI mirrors a similar biphasic motor response to TBI in a rodent model of TBI where rotarod performance was decreased at 2 and 30 days post-TBI, but not at 7 days post-TBI [83].

## dCHI reduces and fragments sleep

Sleep–wake disturbances after TBI are highly prevalent, occurring in 30% to 70% of TBI patients and consisting of insomnia, hypersomnia, fragmented sleep, and altered sleep architecture (reviewed in [84]). In rodent models of TBI, the most commonly reported sleep phenotypes are increased total sleep [85,86–89] and increased sleep fragmentation [85,86,88,90–92].

To test whether sleep is impaired in our TBI model, flies were individually loaded into *Drosophila* Activity Monitors immediately after TBI induction (5 strikes). Flies that died during the behavior runs were included in our analysis. The first 3 days post-TBI, sleep was reduced in TBI-treated flies compared to sham-treated controls ($p < 0.001$, $t$ test with Bonferroni correction; Fig 3C). Also, sleep was fragmented, as seen by a decrease in the length of an average sleep bout ($p < 0.001$, $t$ test with Bonferroni correction; Fig 3D) and an increase in the total number of sleep bouts ($p < 0.01$, $t$ test with Bonferroni correction; Fig 3E). Brief awakenings, a measure of sleep depth [46] were increased ($p < 0.001$, $t$ test with Bonferroni correction; Fig 3F). These sleep phenotypes were not evident after 4 days (Fig 3C–3F). Together, these results show that sleep in flies is decreased and fragmented in the first few days after TBI but that it returns to baseline after 4 days.

To test whether there is a difference in sleep architecture between flies that survived the 7-day experiment and flies that died during the run, we analyzed sleep in both groups separately. Sleep in survivors is decreased (S3A Fig), mostly due to a decrease in night time sleep that persists for 4 days post-TBI (S3A2 Fig). Sleep bout length is decreased (S3B Fig), and the number of sleep bouts is increased (S3C Fig), indicating fragmented sleep. Wake activity was not changed (S3D Fig). On the other hand, the effect of TBI on sleep in flies that died during the experiment is much more limited (S4 Fig).

## dCHI acutely activates the innate immune response

TBI in *Drosophila* affects glial morphology and function [42], including increased blood–brain barrier permeability [93]. Likewise, glia are activated after axonal injury in flies and facilitate clearance of damaged axons [94]. To test whether dCHI could alter glial gene expression, we used TRAP-seq. TRAP-seq allows for a cell type–specific analysis of the all mRNAs that ribosome associated and thus are potentially being translated [49,95]. We used the pan-glial driver *repo-GAL4* [96] to drive *UAS-GFP::RpL10A*, a GFP-tagged version of a ribosomal protein [49]. Head RNA was isolated at 1, 3, and 7 days after TBI induction and compared them to sham-

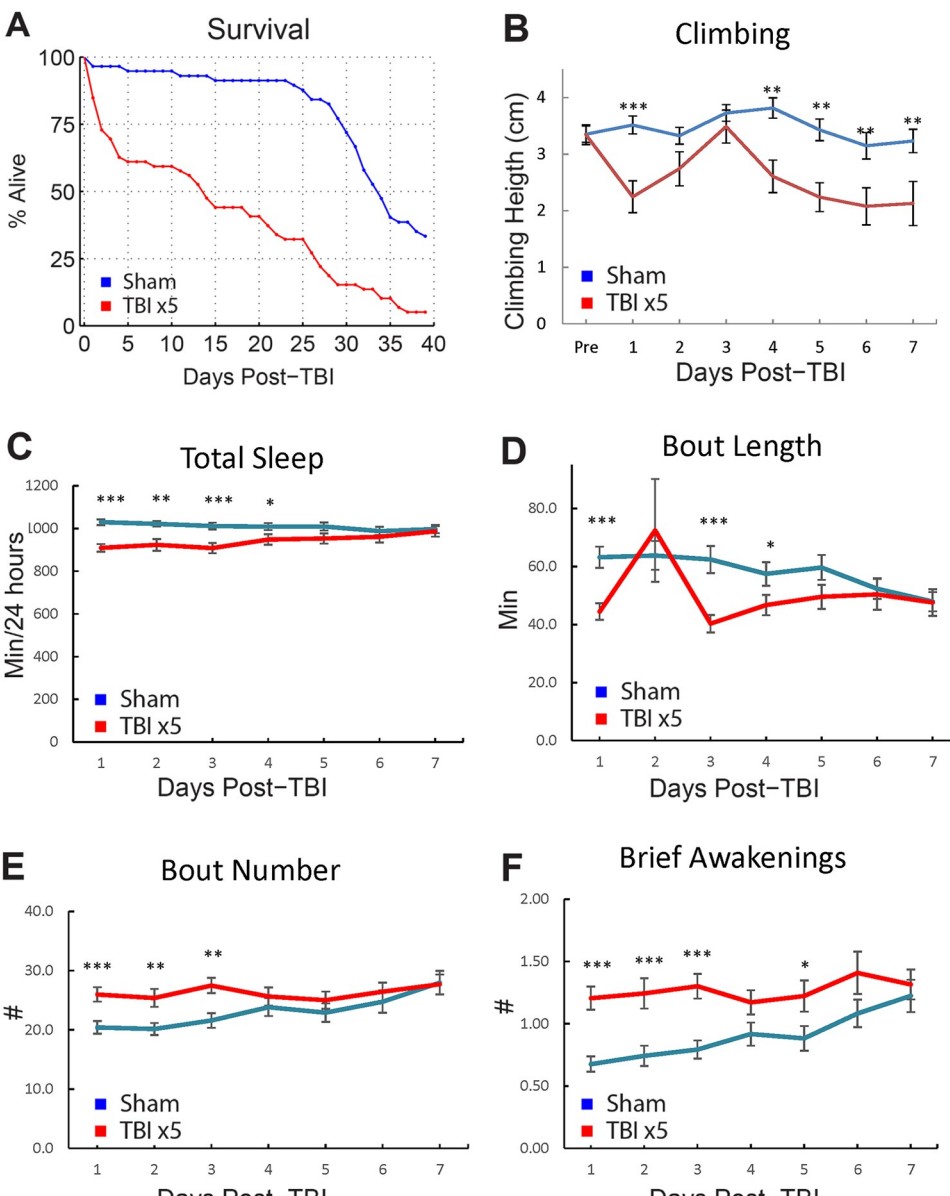

**Fig 3. Long-term effects of TBI on mortality, climbing, and sleep architecture. (A)** Kaplan–Meier estimates of survival functions in TBI-treated flies and sham-treated controls. TBI (5 strikes) was induced in male w1118 flies ($n = 59$), and post-TBI survival was compared to survival in sham-treated controls ($n = 57$) using a log-rank test. TBI results in a significant decrease in survival rate ($p < 0.001$). Around 50% of the TBI group was deceased 14 days after TBI induction, while 50% of the sham-treated controls had died 34 days after the start of the survival assay. **(B)** Climbing behavior was tested in male w1118 flies, after which TBI was induced ($n = 30$). Climbing behavior was subsequently tested for 7 days after TBI and compared to sham-treated controls ($n = 30$). Climbing impairments recover on post-TBI days 2 and 3, followed by a relapse on days 4–7. **(C)** Sleep architecture was quantified in male flies up to 10 days after TBI induction ($n = 96$) and sham-treated controls ($n = 84$). TBI induction resulted in **(C)** decreased total sleep for up to 3 days post-TBI. **(D, E)** More fragmented sleep (decreased bout length, increased bout number) and **(E)** increased brief awakenings, suggesting lighter sleep. *** $p < 0.001$, ** $p < 0.01$ by $t$ tests with Bonferroni correction. Error bars indicate SEM. All figure-related data are located in S3 Data. TBI, traumatic brain injury.

treated controls. Ribosome-associated mRNAs were isolated by immunoprecipitation of the RpL10A-GFP. For each time point, 3 replicates ($n$ = approximately 200 male fly heads/replicate) were collected. Gene expression levels were determined using Kallisto-derived estimated

counts of glial TRAP-seq data collected at 1, 3, or 7 days after dCHI. Due to technical issues with one of day 7 TBI replicates, it was removed from further analysis (see Methods and S5 Fig).

To validate the method, we assessed enrichment of known glial genes in the anti-GFP immunoprecipitate (IP) relative to the input of whole head mRNAs (Fig 4A). Expression levels for non-glia inputs were compared to glia-positive controls and show that glia-specific genes are enriched, including *astrocytic leucine-rich repeat molecule* (*alrm*) [29]), which is expressed highly in astrocytes, *gliotactin*, a transmembrane protein on peripheral glia and *reversed polarity* (*repo*) [96]. Expression of neuron-specific genes embryonic lethal abnormal visual system (elav) and synaptobrevin (nsyb) was not significantly altered (Fig 4A).

We initially looked at genes that are differentially expressed between dCHI and sham-treated controls. Log$_2$ fold change shows how gene expression levels change after dCHI, compared to sham-treated controls. *P* values are adjusted using the Benjamini–Hochberg procedure to correct for false discovery rates [97]. We first noted that the number of differentially expressed genes on day 1 post-dCHI vastly outnumber those evident on day 3 or day 7 (S1–S3 Files). In general, the number of up-regulated genes is higher than the number of down-regulated genes (Fig 4B, S6 Fig). The differentially regulated genes are qualitatively different as well, with little overlap among differentially expressed genes on post-TBI days 1, 3, and 7 (Fig 4C). Only one gene, CG40470, was down-regulated at all 3 time points.

By plotting the average gene levels in the control group against those in TBI-treated flies (Fig 4D), we see that TBI results in genes being more strongly up-regulated than down-regulated (more blue dots, further away from unity line). Strongly up-regulated genes include AMPs (Drosomycin (Drs), listericin, vir-1, genes involved in proteolysis (alphaTry, yip7), and autophagy (MMP1) (Fig 4D). Also, several members of the Turandot (Tot) family are up-regulated after TBI (TotA, TotC, TotM, TotX; Fig 4D). These genes are part of a broad stress response in *Drosophila* and are up-regulated after exposure to mechanical stress, heat, UV, bacteria, oxidative stress, and dehydration [98]. Turandot A (TotA) is strongly induced by bacterial challenge as well as exposure to mechanical pressure, dehydration, and oxidative stress [98]. We also find that several genes with poorly understood functions are strongly up-regulated after TBI (whatever (whe), la costa (lcs); Fig 4D).

The biological processes that are elevated 24 hours after TBI can be roughly grouped in 3 different categories: immune-related, proteolytic/protein folding, and stress response processes. The majority of processes are part of the immune response, including innate immune responses, humoral immune responses, and different classes of AMPs. *Drosophila* AMPs can be grouped into 3 families based on their main biological targets, gram-positive bacteria (Defensin), gram-negative bacteria (Cecropins, Drosocin, Attacins, Diptericin), or fungi (Drosomycin, Metchnikowin) [99]. Most AMP genes were shown to be present in a TRAP-seq analysis of *Drosophila* astrocytes, a glial subset [100]. dCHI results in increased DE of many AMPs, including Attacins, Cecropins, and Diptericins as well as Drosocin, Drosomycin, and Metchnikowin (Fig 5, blue bars). These AMPs are regulated by the Toll, Imd, and JAK–STAT pathways [101]. Previous *Drosophila* TBI models showed an increase in 3 AMPs after TBI induction, Attacin-C, Diptericin B, and Metchnikowin [40,41], but due to the nonspecific nature of the impact, it is uncertain whether they are caused by TBI or by other types of injury. Enriched products of the antiviral and antibacterial JAK–STAT cascade are Listericin, an antibacterial protein, and vir-1, a marker of the induction of an antiviral response. In *Drosophila*, many proteases are involved in the immune response, including the activation of the Toll ligand Spätzle, which is under the control of a serine protease cascade [102]. These proteolytic cascades play a crucial role in innate immune reactions because they can be triggered more quickly than immune responses that require altered gene expression [103]. Expression of

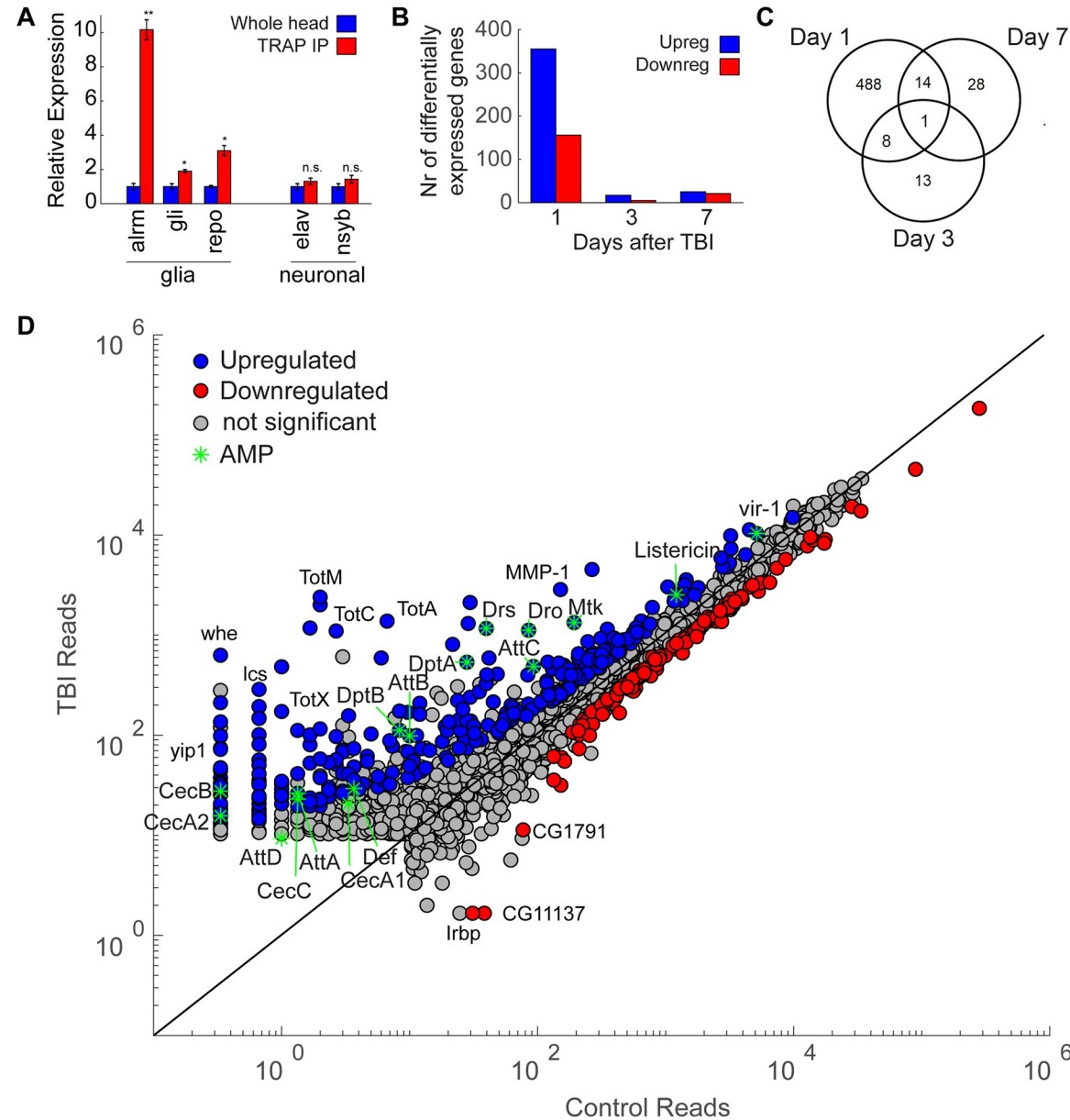

**Fig 4. TBI causes changes in gene expression. (A)** Expression levels for whole head inputs are compared to glia-specific TRAP IP and show that glia-specific genes are enriched, including alrm (astrocyte-specific), gli (expressed in subperineurial glia), and repo (pan-glial). Expression of neuron-specific genes elav and nsyb were not significantly altered. Data are normalized for whole head inputs (3 replicates for both whole head inputs and TRAP IP, ** $p < 0.01$, * $p < 0.05$, $t$ test). Error bars show SEM. **(B)** TBI induction results in a large number of differentially expressed genes in glia cells. Around 356 genes are up-regulated, and 156 are down-regulated 1 day after TBI induction. Three days after TBI, gene expression is almost back to baseline, with only 17 genes that are up-regulated and 5 genes that are down-regulated. Seven days after TBI differential gene expression, 24 genes are up-regulated and 22 are down-regulated. **(C)** Venn diagram showing the overlap of differentially expressed genes (up-regulated + down-regulated) on 1, 3, and 7 days after TBI induction. **(D)** Scatter plot for glial genes where average reads in the control condition are plotted against average reads 24 hours after TBI (blue dots, $\log_2$fold change $\geq 0.6$, Benjamini-adjusted $p < 0.1$) or down-regulated (red dots, $\log_2$fold change $\leq -0.6$, Benjamini-adjusted $p < 0.1$) 24 hours after TBI induction. AMPs are indicated with green asterisks. Genes with average reads <10 in both control and TBI condition were excluded. All figure-related data are located in S4 Data. alrm, astrocytic leucine-rich repeat molecule; AMP, antimicrobial peptide; elav, embryonic lethal abnormal visual system; gli, gliotactin; nsyb, synaptobrevin; repo, reversed polarity; TBI, traumatic brain injury; TRAP IP, TRAP immunoprecipitated.

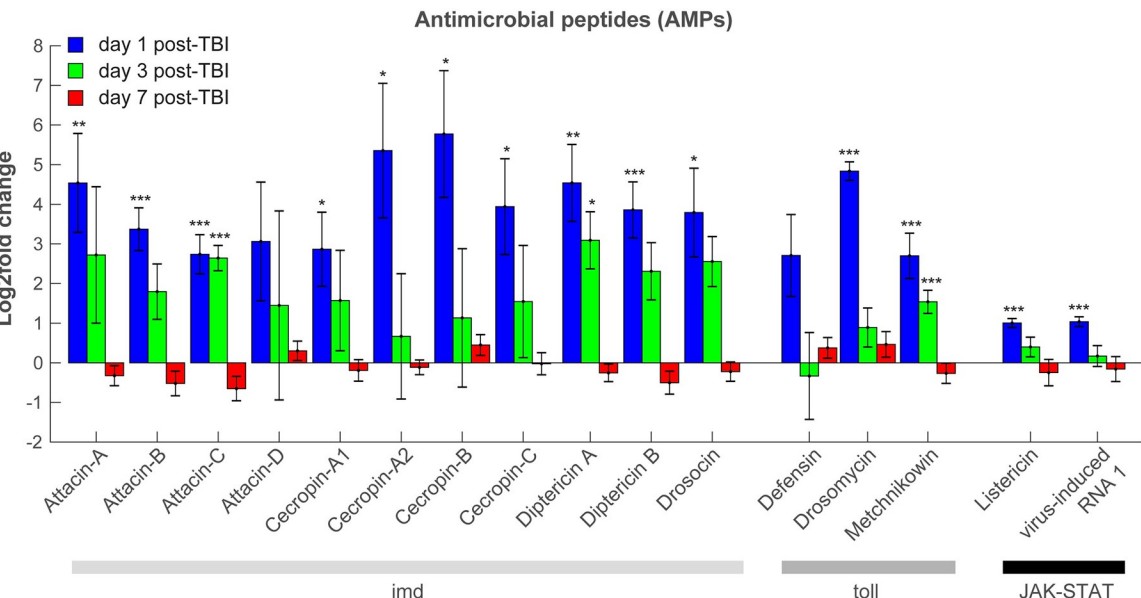

**Fig 5. TBI activates a broad innate immune response.** Twenty-four hours after TBI, genes encoding most *Drosophila* AMPs are enriched in glia cells. These include antibacterial, antifungal, and antiviral peptides that are regulated by the Toll, Imd, and JAK–STAT pathways. Three days after TBI (green), all but 3 AMPs (AttC, DptA, Mtk) have returned to control levels. Seven days after TBI (red), no AMPs are up- or down-regulated. Gene expression levels are considered to have changed significantly if the log2fold change $\geq |0.6|$ and the Benjamini-adjusted $p$-value is $<0.1$. *** adj $p < 0.001$, ** adj $p < 0.01$, * adj $p < 0.1$. Error bars indicate SEM. All figure-related data are located in S4 Data. AMP, antimicrobial peptide; Imd, Immunodeficiency; JAK–STAT, Janus Kinase protein and the Signal Transducer and Activator of Transcription; TBI, traumatic brain injury.

serine proteases that regulate Toll activation (Spirit, Grass, SME, Spheroid, Sphinx 1/2 [104]) were not significantly altered in our assay (S1 File). However, Späztle, a ligand for the Toll pathway that binds to Toll to activate the pathway [105], was up-regulated (1.03 log$_2$ fold change, adj $p$ = 4.69E-06). Activation of the immune response gradually dies down, as only 3 AMPs are up-regulated 3 days after TBI. Seven days after TBI, all AMPs have returned to baseline levels (Fig 5).

In addition to immune gene activation, transcriptomics also uncovered novel pathways and potential treatment targets that mediate TBI effects [106]. A second category of enriched biological processes after TBI are proteases. For example, we detected strong up-regulation of matrix metalloproteinase-1 (MMP-1, 1.79 log$_2$ fold change, adj $p$ = 4.62E-23; Fig 4D, S1 File). MMPs are a family of endopeptidases that have diverse physiological and pathological functions, including degradation of extracellular matrix and regulation of cytokines/chemokines [107]. MMP-1 is induced in *Drosophila* ensheathing glia responding to axonal injury and is required for glial clearance of severed axons [35]. Detecting MMP-1 in our dCHI assay is further evidence that TBI damages axons and triggers glia-mediated neuroprotective responses.

The third main category of enriched genes 24 hours after TBI are stress response genes, genes that are up-regulated in response to a variety of stressful stimuli, including heat, oxidative, metabolic, and chemical stress [98,108], including the Tot genes and heat shock proteins 22, 23, 26, 27, 68, 70Bc, and lethal-2 as well as genes involved in oxidative stress. Heat shock protein 27 was previously implicated in response to bacteria and fungi as well [109].

In our model, TBI causes sleep to be reduced, fragmented, and less deep (Fig 3C–3F), a phenotype that persists for 3 days after TBI induction. Two prominent sleep-regulating genes have altered expression levels 24 hours after TBI (S1 File). Dopamine transporter (*DAT*, also known as *fumin*), a dopamine transporter that mediates uptake of dopamine from the synaptic cleft, is

down-regulated ($-0.74$ $\log_2$ fold change, $p = 0.021$). Loss of *DAT* increases extracellular dopamine and is associated with increased activity and decreased sleep [110–112]. *Pale* (*ple*), a tyrosine hydroxylase that drives synthesis of wake-promoting dopamine [81], is up-regulated ($0.63$ $\log_2$ fold change, $p = 0.018$). *Pale* was previously shown to be activated in response to wounding in *Drosophila* embryos and larvae [113]. *Pale* and *DAT*/*fumin* levels are not changed on post-TBI days 3 and 7 (S2 and S3 Files). Thus, we hypothesize down-regulation of *DAT*/*fumin* in combination with up-regulation of *pale* may underlie TBI changes to sleep due to increased dopamine levels.

Three days after TBI induction, there are few significant differences in gene expression between TBI-treated flies and sham-treated controls (Fig 4B, S7A Fig, S2 File). Whereas there are 512 genes with altered expression 24 hours post-TBI (almost 400 of those are up-regulated), after 3 days, there are only 22 genes with altered expression levels (Fig 4B). Interestingly, this low level of glial activation at day 3 post-TBI coincides with a climbing behavior returning back to control levels (Fig 4B). At 3 days post TBI, several AMPs remain strongly up-regulated (AttC, Mtk, Dpt; Fig 5, S7A Fig).

Seven days after TBI induction, there is more variability in gene expression (S5E and S5F Fig, S3 File). There seems to no activation of the immune response, as all AMPs have returned to baseline levels (Fig 5, red bars). There is only one gene that is persistently down-regulated on days 1, 3, and 7 (Fig 4C). CG40470's function is unknown, although roles in proteolysis and peptide catabolic processes have been inferred [114].

## Post-TBI behavioral phenotypes are NF-κB dependent

The NF-κB family of transcription factors plays a central role in the regulation of inflammatory gene expression [115], cell survival, and neuronal plasticity [116]. NF-κB is activated in neurons and glial cells after injury and has been linked to both neurodegenerative and neuroprotective activities [116,117]. NF-κB mediates activation of glial cells [118] and inflammation [119]. TBI causes an increase in NF-κB in rodents [120–122], where it has a neuroprotective effect in a closed head model of TBI [83]. In *Drosophila*, NF-κB is a crucial component of both the Toll and the Imd pathways, where different isoforms are required for the expression of different AMPs [123]. Changes to sleep architecture after injury/infection in *Drosophila* require NF-κB *Relish* [124]. In *Drosophila*, overexpression of NF-κB or AMPs in glia cells causes neurodegeneration [38,39,125,126].

To test the role of NF-κB in TBI-induced mortality and behavioral impairments, we induced TBI in the NF-κB *Relish* null mutant (w1118; Rel[E20], Bloomington #9457) and measured its effects on post-TBI survival, climbing behavior, and sleep. Five consecutive strikes to the head resulted in strongly increased mortality (log-rank test on Kaplan–Meier survival curves, $p < 0.001$), where over 50% of the NF-κB mutants had died 3 days after TBI induction (Fig 6A). In the background strain ($w^{1118}$), 50% mortality for 5 strikes occurs at approximately 14 days post-TBI (Fig 3A). In sham-treated NF-κB controls, 50% mortality occurs after 35 days (Fig 6A), which is very similar to mortality in the sham-treated $w^{1118}$ background strain (Fig 3A).

To test whether TBI has a much stronger effect on NF-κB mutants, we tested climbing behavior daily for 7 days after TBI induction. However, there was no difference between TBI-treated flies and untreated controls on any day (Fig 6B). This is different from wild-type flies, which show considerable impairment to climbing 24 hours post-TBI, which then reverts back to normal and is followed by a relapse on days 5 to 7 (Fig 3B). Both TBI-treated and sham-treated controls show a gradual decrease in total sleep over 10 days posttreatment. However, there are no differences in total sleep (Fig 6C) or in any other metrics of sleep architecture

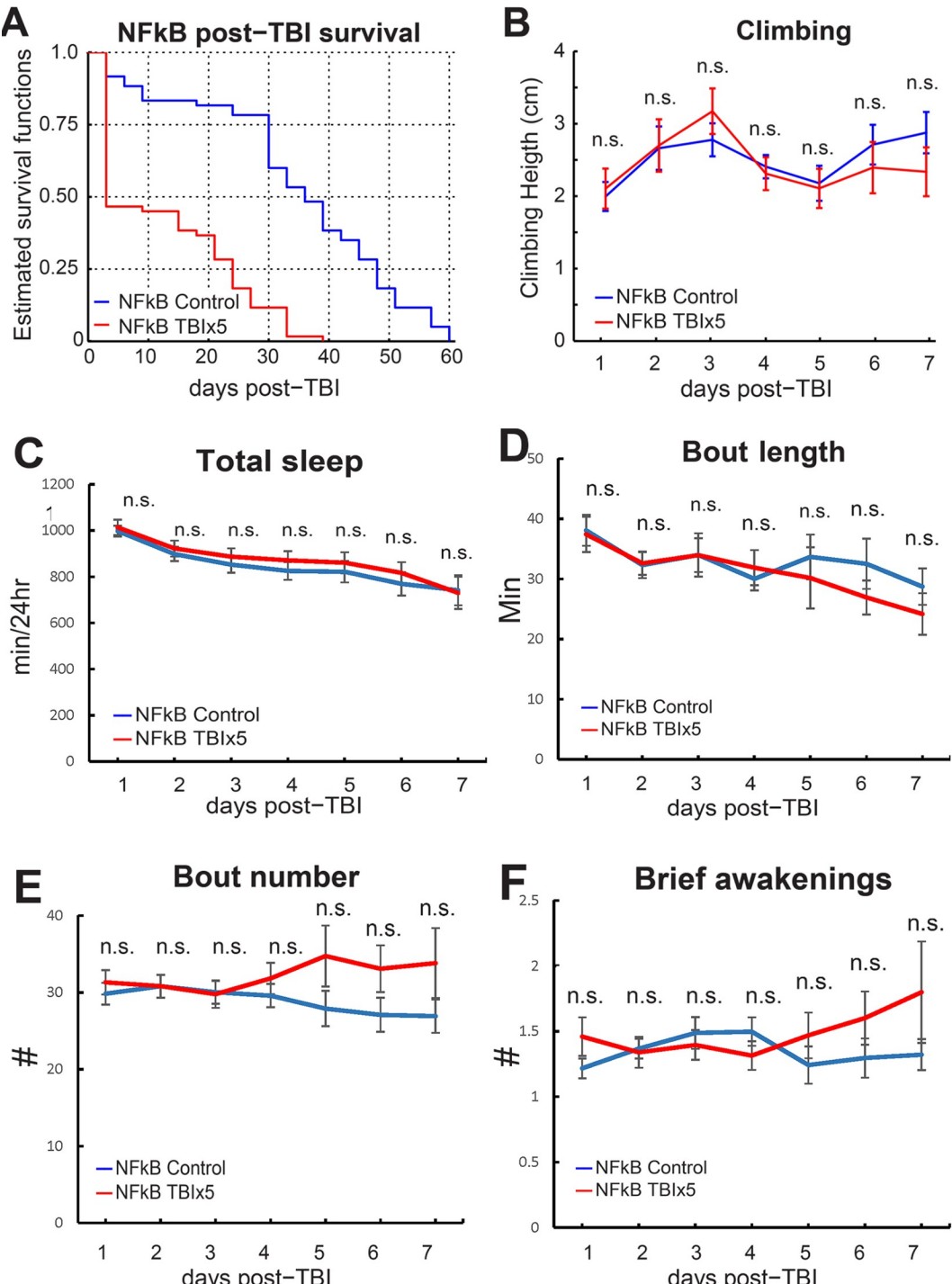

**Fig 6. Post-TBI behavioral phenotypes are NF-κB dependent. (A)** Kaplan–Meier estimates of survival functions in TBI-treated NF-κB null mutants and sham-treated controls ($n = 60$/group) show that TBI significantly reduces life span (log-rank test, $p < 0.001$). **(B)** The effect of TBI on climbing behavior was tested in male NF-κB null mutants and sham-treated controls ($n = 32$/group for 7 consecutive days after TBI induction. There was no significant difference between both groups (n.s., $t$ tests with Bonferroni correction. **(C)** TBI does not impair sleep in NF-κB null mutants, compared to sham-treated controls. TBI does not fragment sleep as average bout length **(D)** and bout number **(E)** are not changed. **(F)** Brief awakenings, a measure of sleep depth, are unchanged after TBI. ($n = 57$ controls, 49 in TBI group, n.s.; $t$ tests with Bonferroni correction). Error bars indicate SEM. All figure-related data are located in S5 Data. NF-κB, nuclear factor kappa B; n.s., not significant; TBI, traumatic brain injury.

(brief awakenings, bout length, bout number, wake activity; Fig 6D–6F). These results suggest that the NF-κB–dependent immune response facilitates survival after TBI but that impairments in sleep and climbing behavior are consequences of an immune-dependent injury mechanism.

The TBI-induced innate immune response can be either beneficial [14] or harmful [15–18]. TBI in *Drosophila* results in strong up-regulation of many AMPs (Fig 5). To test whether they, in addition to their antimicrobial effects, also confer beneficial or detrimental effects on TBI survival, we used RNAi-mediated glial knockdown of AMPs. Unfortunately, glial knockdown of single AMPs either failed to consistently impact TBI survival (S8 Fig), suggesting that AMP functions may be redundant. Indeed, cooperation of AMPs has been demonstrated to enhance their microbicidal activity in *Drosophila* [127].

To address AMP classes, we assessed flies where entire classes of immune-inducible AMPs have been knocked out [127]: flies lacking the primarily *Defensin* (Group A), flies lacking *Drosocin*, as well as *Diptericin A and B* and *Attacin A-D* (Group B), and flies lacking 2 antifungal peptide genes *Metchnikowin* and *Drosomycin* (Group C). These 3 groups were then combined to generate flies lacking AMPs from groups A and B (AB), A and C (AC), or B and C (BC) as well as a group lacking all 3 groups (ABC or Δ*AMPs*) [127]. In all of these classes, *Cecropins* are still present, after a recombination event reintroduced a wild-type Cecropin locus [127].

We first tested whether loss of all 14 AMP classes (ABC) affects TBI survival. These flies are highly susceptible to numerous infections [127]. As expected, flies lacking all 3 classes or any 2 classes show strongly increased TBI-induced mortality or any 2 classes (Fig 7A; log-rank test on Kaplan–Meier survival curves, $p = 0.0006$; Fig 7B, AB, $p = 0.019$; Fig 7C, AC, $p = 0.0008$; Fig 7D, BC, $p = 0.012$). Surprisingly, flies only lacking class A show increased survival (Fig 7E, A, $p = 0.026$), suggesting that Defensin up-regulation is detrimental rather than beneficial for TBI survival. Flies lacking either class B (Fig 7F) or class C (Fig 7G) show no effect on survival, even though loss of both classes simultaneously increases mortality (Fig 7D). Likewise, flies lacking classes A and B or AC show increased mortality, even though class A flies show increased survival and classes B and C show no significant increase in survival, suggesting that these classes interact in a nonsynergistic manner. Together, these results indicate that most AMPs confer survival benefits but that some immune activation is detrimental. Regardless, these results reveal that AMPs are crucial mediators of TBI effects on survival. To test whether flies lacking all AMP classes (ABC) show other TBI phenotypes, we tested climbing in a negative geotaxis assay and sleep and found that, as in iso31 controls, climbing in ABC flies is decreased after TBIx5 (S9A Fig). Where the control line shows impaired sleep after TBIx5 (S9B Fig), ABC flies show increased, not decreased, sleep after TBI (S9C Fig), further underscoring their role in mediating TBI behavioral effects.

## Discussion

We have developed a straightforward and reproducible *Drosophila* model for closed head TBI where we deliver precisely controlled strikes to the head of individually restrained, unanesthetized flies. This TBI paradigm is validated by recapitulating many of the phenotypes observed in mammalian TBI models, including increased mortality, increased neuronal cell death, impaired motor control, decreased/fragmented sleep, and hundreds of TBI-induced changes to the transcriptome, including the activation of many AMPs, indicating a strong activation of the immune response. These results set the stage to leverage *Drosophila* genetic tools to investigate the role of the immune response as well as novel pathways in TBI pathology.

Our single fly paradigm is a more valid *Drosophila* model for TBI that circumvents the lack of specificity of currently available models [39,41] or the use of anesthesia [42]. Both previous

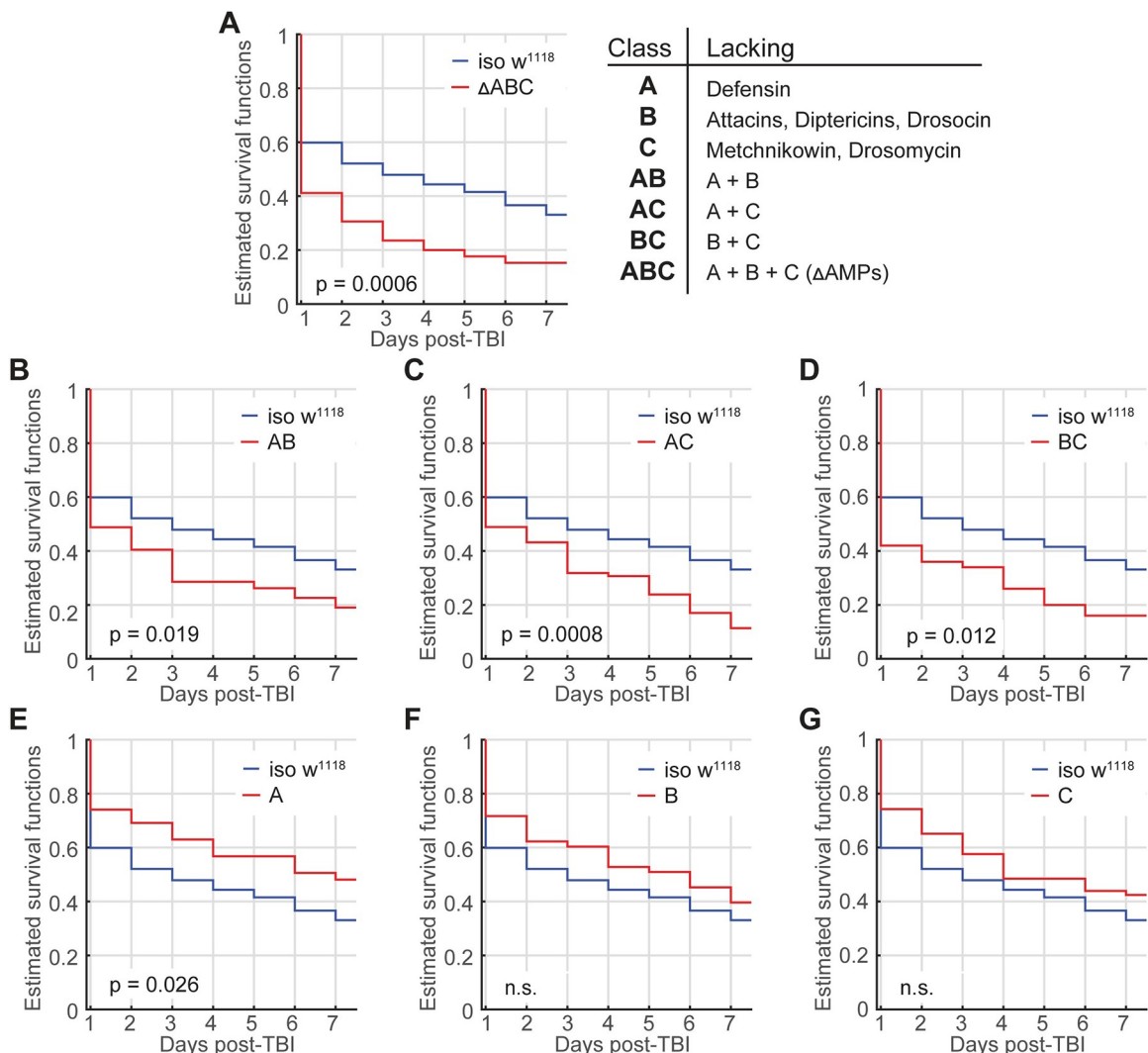

**Fig 7. Nonsynergistic effects of AMPs on TBI mortality. (A)** Flies lacking all classes on AMPs (ΔAMP) are highly susceptible to TBI. Likewise, flies lacking most classes of AMP show increased mortality after TBI. **(B-D)** Flies lacking only Defensin (**E**, class A) show increased survival after TBI, while loss of Attacins, Diptericins, and Drosocin (**F**, class B) and the loss of antifungals (Metchnikowin and Drosomycin) (**G**, class C) have no significant effect on survival. n.s = not significant (log-rank test); *n* = 55–85. All figure-related data are located in S6 Data. AMP, antimicrobial peptide; TBI, traumatic brain injury.

assays induce TBI by subjecting the whole fly to trauma, which makes it hard to distinguish whether observed phenotypes are a due to TBI or a consequence of internal injuries. A recently published method [128] uses a pneumatic device to strike an anesthetized fly's head. This method is an improvement of earlier assays and results in increased mortality in a stimulus strength–dependent manner. However, it only shows a modest reduction in locomotor activity, without demonstrating any other TBI-related phenotypes such as neuronal cell death or immune activation. The dependence on $CO_2$ anesthesia further impairs the usefulness of this assay, as prolonged behavioral impairments in *Drosophila* occur even after brief exposure to $CO_2$ anesthesia [129]. Additionally, anesthetics that are administered either during or shortly after TBI induction can offer neuroprotective effects and alter cognitive, motor, and histological outcomes in mammalian models of TBI [130–132] as well as affecting mortality in a whole body injury model in flies [133]. Our *Drosophila* model allows us to study how TBI affects

behavior and gene expression without the confounding effects of anesthesia, making it a more valid model for TBI that occurs under natural conditions.

The force used in our study (8.34 N) is higher than the force used in the HIT assay (2.5 N) [40]. When designing our TBI paradigm, we tested several commercially available solenoids for their ability to induce TBI and used the one that gave the best results. We may need a higher force because brain damage is caused by the direct impact of the solenoid to the fly head, where the fly head moves with the solenoid (S1 Movie) rather than full body injury or compression injuries used in the other *Drosophila* TBI assays. Although we cannot exclude that the neck is not damaged in our assay, we do observe cell death in the central brain and observe significant changes in glia after TBI, suggesting that TBI does occur.

In this study, we also elucidate, in an unbiased manner, the genomic response to TBI. Glial cells play an important role in immune responses in both mammals and *Drosophila* (reviewed in [33]), and changes to glial morphology and function were reported in earlier *Drosophila* TBI models [42,93]. Until now, profiling TBI-induced changes in gene expression have either been limited to a small number of preselected genes in both mammals [134,135] and *Drosophila* [40,41] or focused on whole brain tissue rather than individual cell types [136,137]. Using TRAP in combination with RNA-seq, we validate previously reported up-regulation of Attacin-C, Diptericin-B, and Metchnikowin [40]. Additionally, we detected an acute, broad-spectrum immune response, where AMPs and stress response genes are up-regulated 24 hours after TBI. These include antibacterial, antifungal, and antiviral peptides as well as peptides from the Tot family, which are secreted as part of a stress response induced by bacteria, UV, heat, and mechanical stress [98]. Although an increase in the heatshock protein 70 family of stress response genes was reported earlier [42], we only detected a significant glial up-regulation in Hsp70BC (logfold change: 2.02, adj. $p$ = 0.031; S1 File).

Three days after TBI, only Attacin-C, Diptericin A, and Metchnikowin are up-regulated. Seven days after TBI, AMPs or stress response genes are not detectably up-regulated. These findings match reports in mammalian TBI models, where inflammatory gene expression spikes shortly after TBI but mostly dies down during subsequent days [134,138]. Using CRISPR deletions of AMP classes, we demonstrate that most AMPs not only protect against microbes but are also crucial in promoting survival after TBI. The exception is Defensin, as loss of this peptide increases survival, indicating that the *Drosophila* innate immune response to TBI can have both beneficial and detrimental effects. While loss of AMPs may render flies more susceptible to TBI, we favor the hypothesis that AMP induction after TBI actively plays a role in mediating TBI effects.

Besides validating our *Drosophila* model with the detection of a strongly up-regulated immune response, we detected several novel genes among the total of 512 different glial genes that were either up- or down-regulated after TBI. Immune and stress response only make up 157 out of 512 differentially expressed glial genes. Genes involved in proteolysis and protein folding are a prominent portion (85/512) of these differentially expressed genes, yet their role in TBI is poorly understood. These results demonstrate that there are other candidate pathways that may modulate recovery, and *Drosophila* can be used as a first line screen to test their in vivo function and to disentangle beneficial from detrimental responses.

We have successfully applied in vivo genetics to identify in vivo pathways important for TBI. Here, we demonstrate that loss of master immune regulator NF-κB results in increased mortality after TBI, yet it protects against TBI-induced impairments in sleep and motor control. These findings align with previous reports showing links between sleep and the immune response in flies [139] where NF-κB is required to alter sleep architecture after exposure to septic or aseptic injuries [140]. It will be of interest to determine if NF-κB is required for TBI-induced cell death. One possibility is that sleep impairments can be a side effect of

melanization, an invertebrate defense mechanism that requires dopamine as melanin precursor [141]. If dopamine is up-regulated to create more melanin, decreased sleep would be a side effect. Consistent with this hypothesis, we observe changes in *fumin* and *pale*, which likely result in increased dopamine levels.

However, the role of sleep after injury is complex. Two recent studies demonstrated that sleep is increased after antennal transection and facilitates Wallerian degeneration and glia-mediated clearance of axonal debris [94,142], suggesting that different types of injury have different effects on sleep. Interestingly, sleep disturbances can increase the up-regulation of immune genes [139,143]. Thus, it is possible that decreased sleep after TBI contributes to survival by stimulating the immune response. We find some support for this hypothesis in the difference in TBI-induced changes to sleep in flies that survive 7 days of TBI versus flies that die within 7 days after TBI, where the survivors sleep significantly less for 4 days post-TBI (S3 Fig) and dying flies sleep is nearly unaffected (S4 Fig). Additionally, immune response genes are up-regulated for up to 3 days after TBI (Fig 5), which correlates with our observed sleep impairments (Fig 3C). Also, the engulfment receptor Draper, which mediates Wallerian degeneration, is not up-regulated in our glial TRAP-seq data (log2fold change = 0.26, adj. *p* = 0.74; S1 File), suggesting that Wallerian degeneration, and its accompanying increase in sleep, is not part of the response to dCHI.

TBI results in impaired climbing behavior that persists for up to 7 days (Fig 3B), yet impairments to sleep disappear after a few days (Fig 3C). Recently, it was shown that TBI through head compression results in impaired memory, as quantified through courtship conditioning [42], indicating that TBI also results in persistent memory defects.

Recently, it was shown that repressing neuronal NF-κB in a mouse model of TBI increases post-TBI mortality, as in our studies, without reducing behavioral impairments [83], suggesting that nonneuronal NF-κB could underlie behavioral impairments after TBI. We demonstrate that behavioral responses to TBI (for example, sleep and geotaxis) are abolished in mutants of the transcription factor NF-κB *Relish*, which plays a central role in regulating stress-associated and inflammatory gene expression in both mammals [116,144] and flies [145]. Nonetheless, *Relish* null mutants show increased mortality after TBI, but none of the behavioral impairments observed in wild-type flies, indicating that these impairments might be a side effect of immune activation rather than direct injury. The demonstration of an in vivo role for TBI-regulated genes will be important for defining their function.

In summary, our dCHI assay recapitulates many of the physiological symptoms observed in mammals, demonstrating that fruit flies are a valid model to study physiological responses to TBI. We demonstrate both a potent induction of immune pathways and a requirement for an immune master regulator in mediating TBI effects on behavior. Our model now provides a platform to perform unbiased genetic screens to study how gene expression changes after TBI in unanesthetized, awake animals result in the long-term sequelae of TBI. These studies raise the possibility of rapidly identifying key genes and pathways that are neuroprotective for TBI, thereby providing a high-throughput approach that could facilitate the discovery of novel genes and therapeutics that offer better outcomes after TBI.

## Supporting information

**S1 Fig. TBI results in immediate locomotion defects in a dose-dependent manner. (A)** Representative position traces for single flies during the first 4 hours immediately after TBI onset, for controls as well as flies in the TBIx1, TBIx5, and TBIx10 conditions. **(B)** TBI resulted in a dose-dependent number of flies being immobile immediately after TBI onset (*** *p* < 0.001 chi-squared test). **(C)** It took these flies second (TBIx1) to minutes (TBIx5 and TBIx10) to

become active (* $p < 0.05$, *** $p < 0.001$, $t$ test). **(D)** Locomotor defects (circling, slow walking, sideways walking, backwards walking, and jumping) occurred shortly after TBI onset, in a dose-dependent manner. Locomotor defects only were only observed in flies that were immobile after TBI (*** $p < 0.001$ chi-squared test). **(E)** Walking speed was reduced in all 3 groups during the first hour post-TBI, but the TBIx1 and TBIx5 groups had recovered by the second hour. Walking speed remained impaired for all 4 hours in the TBIx10 group. **(F)** Overall activity (% of time active) was significantly reduced in the TBIx5 and TBIx10 groups for the first hour after TBI, but unaffected in the TBIx1 group (* $p < 0.05$, ** $p < 0.01$, *** $p < 0.001$, $t$ test). $n = 20$–$24$ per TBI group, 32 controls. Error bars indicate SEM. Movie extracted data can be found in S1 Data. TBI, traumatic brain injury.
(TIF)

**S2 Fig. Early deaths do not fully account for TBI-induced increase in mortality.** To test whether increased mortality due to TBI can be explained by early deaths, we set mortality to zero cumulatively for the first 2 weeks post-TBI. In all instances, we see significantly increased mortality in the TBI-treated group (log-rank test), indicating that the observed increase in mortality is not due to early deaths only. All figure-related data are located in S2 Data. TBI, traumatic brain injury.
(TIF)

**S3 Fig. Sleep is decreased in TBI survivors.** To test whether sleep affect flies that survive our 7-day sleep experiment differently than flies that die during this experiment, we split our sleep data in survivors and dying flies. Sleep data for survivors is shown here. **(A1–3)** Total sleep during the day, night and total sleep shows that post-TBI is mostly decreased during the night, for up to 4 days post TBI. **(B1–3)** Average sleep bout length was modestly reduced, and **(C1–3)** sleep bout numbers were increased, suggesting that sleep is both decreased and fragmented for the first 3 days after TBI. **(D1–3)** Wake activity was not affected by TBI during the first 7 days post-TBI. $n = 67$ sham-treated and 56 TBI flies. *** $p < 0.001$, ** $p < 0.01$ by $t$ tests with Bonferroni correction. Error bars indicate SEM. All figure-related data are located in S3 Data. TBI, traumatic brain injury.
(TIF)

**S4 Fig. Sleep is largely unaffected in flies dying within 7 days post-TBI.** To test whether sleep affect flies that survive our 7-day sleep experiment differently than flies that die during this experiment, we split our sleep data in survivors and dying flies. Sleep data for dying flies is shown here. **(A1–3)** Total sleep during the day, night and total sleep shows that post-TBI is increased during the day on days 2 and 3 post-TBI. **(B1–3)** Average sleep bout length was strongly increased during the day, but **(C1–3)** sleep bout numbers were unaffected, suggesting that sleep more consolidated during post-TBI days 2 and 3. **(D1–3)** Wake activity during the day was not affected by TBI during the first 3 days post-TBI, indicating that the observed sleep effect is not due to decreased locomotion. $n = 17$ sham-treated and 40 TBI flies. *** $p < 0.001$, ** $p < 0.01$ by $t$ tests with Bonferroni correction. Error bars indicate SEM. All figure-related data are located in S3 Data. TBI, traumatic brain injury.
(TIF)

**S5 Fig. Relative log expression and normalized data for post-TBI days 1, 3, and 7.** RLE plot of raw and normalized glial expression data. Control (green) and TBI (orange) biological replicates for days 1, 3, and 7 post-TBI. Correction was performed using the UQ normalization method. Due to the high variability in TBI replicate 3 on day 7, this replicate was discarded. RNA-seq data were deposited under accession number GSE164377. RLE, relative log

expression; TBI, traumatic brain injury; UQ, upper-quartile.
(TIF)

**S6 Fig. Glial gene expression heat map.** Panels present clustering of DEGs for day 1 post-TBI. Gene expression level presented as z-scored log2(X+1) transformed values; control replicates in blue, TBI replicates in red. All figure-related data are located in S4 Data. DEG, differentially expressed gene; TBI, traumatic brain injury.
(TIF)

**S7 Fig. Differentially expressed genes in repo-TRAP at 3 and 7 days post-TBI.** Scatter plot for glial genes where average reads in the control condition are plotted against average reads 3 days (**A**) or 7 days (**B**) after TBI (blue dots, log$_2$fold change $\geq 0.6$, Benjamini-adjusted $p < 0.1$) or down-regulated (red dots, log$_2$fold change $\leq -0.6$, Benjamini-adjusted $p < 0.1$) 24 hours after TBI induction. AMPs are indicated with green asterisks. Genes with average reads <10 in both control and TBI condition were excluded. All figure-related data are located in S4 Data. AMP, antimicrobial peptide; TBI, traumatic brain injury.
(TIF)

**S8 Fig. Glia-specific AMP knockdown does not affect post-TBI mortality.** Kaplan–Meier plots for glia-specific RNAi-mediated knockdown of AMPs. Repo>RNAi lines are compared to Repo>RNAi control lines using a log-rank test. n.s. = not significant. (**A**) Attacin-A, (**B**) Attacin-B, (**C**) Attacin-C, (**D**) Cecropin-A, (**E**) Cecropin-B, (**F**) Cecropin-C, (**G**) Diptericin-A, (**H**) Diptericin-B, (**I**) Drosocin, (**J**) Drosomycin, (**K**) Listericin, (**L**) Metchnikowin, (**M**) virus-induced RNA 1. All figure-related data are located in S7 Data. AMP, antimicrobial peptide; TBI, traumatic brain injury.
(TIF)

**S9 Fig. ΔAMP null mutants exhibit decreased climbing and increased sleep after TBI.** (**A**) ΔAMP null mutants show decreased climbing 24 hours after TBI, similar to controls ($n = 50–66$) (**B**) Sleep is decreased in controls 24 hour after TBI (**C**) but is increased in ΔAMP null mutants ($n = 56–95$). $^{***} p < 0.001$, $^* p < 0.05$ by $t$ tests with Bonferroni correction. Error bars indicate SEM. All figure-related data are located in S2 Data. AMP, antimicrobial peptide; TBI, traumatic brain injury.
(TIF)

**S1 Data. Data corresponding to S1 Fig.**
(XLSX)

**S2 Data. Data corresponding to Figs 2, S2 and S9.**
(XLSX)

**S3 Data. Data corresponding to Figs 3, S3 and S4.**
(XLSX)

**S4 Data. Data corresponding to Figs 4, 5, S5, S6 and S7.**
(XLSX)

**S5 Data. Data corresponding to Fig 6.**
(XLSX)

**S6 Data. Data corresponding to Fig 7.**
(XLSX)

**S7 Data. Data corresponding to S8 Fig.**
(XLSX)

**S1 File. Differential gene expression in glial cells between TBI and control, 1 day after TBI.**
Related to Figs 4 and S6. TBI, traumatic brain injury.
(XLSX)

**S2 File. Differential gene expression in glial cells between TBI and control, 3 days after TBI.** Related to Figs 4 and S6. TBI, traumatic brain injury.
(XLSX)

**S3 File. Differential gene expression in glial cells between TBI and control, 7 days after TBI.** Related to Figs 4 and S6. TBI, traumatic brain injury.
(XLSX)

**S1 Movie. TBI procedure in *Drosophila*.** Related to Fig 1. TBI, traumatic brain injury.
(MP4)

**S2 Movie. Mobility and response to tactile stimulus immediately after TBI.** Related to S1 Fig. TBI, traumatic brain injury.
(MP4)

**S3 Movie. Restoration of mobility and stimulus response after TBI.** Related to S1 Fig. TBI, traumatic brain injury.
(MP4)

## Acknowledgments

The authors thank Eugenie Bang for conducting the experiments that assessed TBI susceptibility and sleep and motor impairments in wild-type and NF-κB null flies. The authors thank Dr. Rosemary Braun for supervising the bioinformatic analyses and reviewing the initial draft of this manuscript.

## Author Contributions

**Conceptualization:** Bart van Alphen.

**Data curation:** Bart van Alphen.

**Formal analysis:** Bart van Alphen, Samuel Stewart, Marta Iwanaszko, Fangke Xu, Anujaianthi Ramakrishnan, Shiju Sisobhan, Zuoheng Qin, Bridget C. Lear.

**Funding acquisition:** Bart van Alphen, Ravi Allada.

**Investigation:** Bart van Alphen, Samuel Stewart, Fangke Xu, Keyin Li, Sydney Rozenfeld, Anujaianthi Ramakrishnan, Taichi Q. Itoh, Zuoheng Qin.

**Methodology:** Bart van Alphen, Samuel Stewart, Anujaianthi Ramakrishnan.

**Resources:** Bart van Alphen, Ravi Allada.

**Supervision:** Ravi Allada.

**Validation:** Bart van Alphen.

**Visualization:** Bart van Alphen.

**Writing – original draft:** Bart van Alphen, Ravi Allada.

**Writing – review & editing:** Bart van Alphen, Samuel Stewart, Marta Iwanaszko, Fangke Xu, Keyin Li, Sydney Rozenfeld, Anujaianthi Ramakrishnan, Taichi Q. Itoh, Shiju Sisobhan, Bridget C. Lear, Ravi Allada.

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
