## [Editor Report · Decision Letter 0]

28 Jul 2020

Dear Dr Allada, 

Thank you for submitting your manuscript entitled "Glial immune-related pathways as mediators of closed head TBI effects on behavior and lethality in Drosophila" for consideration as a Research Article by PLOS Biology.

Your manuscript has now been evaluated by the PLOS Biology editorial staff as well as by an academic editor with relevant expertise and I am writing to let you know that we would like to send your submission out for external peer review.

Please re-submit your manuscript within two working days, i.e. by Jul 30 2020 11:59PM.

Kind regards,

The *PLOS Biology* Editors

---

## [Decision Letter · Decision Letter 1]

23 Sep 2020

Dear Dr Allada,

Thank you very much for submitting your manuscript "Glial immune-related pathways as mediators of closed head TBI effects on behavior and lethality in Drosophila" for consideration as a Research Article at PLOS Biology. Thank you also for your patience as we completed our editorial process, and please accept my apologies for the delay in providing you with our decision. Your manuscript has been evaluated by the PLOS Biology editors, an Academic Editor with relevant expertise, and by four independent reviewers.

The reviews are attached below. You will see that the reviewers agree that the paper presents an interesting new model of traumatic brain injury (TBI), however they also raise several concerns and suggest new experiments to strengthen the manuscript. After consulting with the Academic Editor, these are our thoughts regarding the experiments that need to be performed in order for us to consider a revision of the manuscript. Reviewer 1 raises several issues that you should try to address with changes in the text. We find Reviewer 2’s request regarding the separation of animals by how quickly they die reasonable and should be done.This reviewer also asks for new data to determine whether the TUNEL label is marking glia or neurons, but TUNEL may not be compatible with immune-fluorescence. It would be possible to use cell death markers like activated caspase3 with Repo and Elav antibodies, but it could be difficult to perform, thus although we would welcome the results, we won’t make this experiment a requirement for publication. We would like to see the sleep analysis separated by day vs night, but we won’t require TUNEL labelling in relish animals after TBI and checking whether or not effects of immune genes on TBI are specific to TBI or just make the flies sicker, although you should discuss the latest point. Reviewer 3 raises a point that we find critical regarding the control of the genetic background and you must provide the information requested; you should also provide a better description of the behavior and discuss memory deficits, but the other experiments suggested by this reviewer in the major points, while nice, we think they are out of scope for this study. Regarding Reviewer 4’s suggestions, we would like to see an analysis of the genes altered in the TBI assay and you should address all the other points, except for analyzing the mechanisms that lead to cell death.

In light of the reviews, we will not be able to accept the current version of the manuscript, but we would welcome re-submission of a revised version that takes into account the reviewers' comments. We cannot make any decision about publication until we have seen the revised manuscript and your response to the reviewers' comments. Your revised manuscript is also likely to be sent for further evaluation by the reviewers.

We expect to receive your revised manuscript within 3 months. 

**IMPORTANT - SUBMITTING YOUR REVISION**

*Re-submission Checklist*

*Published Peer Review*

*PLOS Data Policy*

*Blot and Gel Data Policy*

Sincerely,

The *PLOS Biology* Editors

Reviewers’ comments

Rev. 1:

The authors describe a method to deliver strikes to the head of anesthetized flies to establish a novel traumatic brain injury (TBI) model. They performed a number of tests to determine the action of TBI on survival, negative geotaxis, sleep and gene expression. Sequence analysis identified an upregulation of a larger number of genes including those involved in the innate immune response which decreases after 7 days post TBI. Only one gene is persistently downregulated on all days tested (CG40470). The authors then tested the role of NF-kB signaling and performed TBI in a Relish null mutant. This showed an increased immediate mortality and moreover indicated that Post-TBI behavioral phenotypes are NF-kB -dependent. In addition, they showed that AMP deficiency has an impact on the TBI survival.

TBI is an important subject to study and work in Drosophila might potentially help to understand human pathology. However, I do have some concerns regarding the brain specific trauma induction, this approach in my view is likely to generate disruptions of (1) nerves running in the neck of the fly and disruptions in the foregut which is also located in the neck. In addition, the forces used in this study (8.34 N) appear quite high in particular when comparing to the forces obtained in the device of Katzenberger et al., (2013, 2 N). This should at least be discussed. And more importantly it must be analyzed whether the neck is not damaged.

The present work provides a number of interesting finding - which however are not entirely new. The Bonini group just published a paper "Dynamic neural and glial responses of a head-specific model for traumatic brain injury in Drosophila" presenting a head-specific TBI model (Saikumar et al., PNAS 2020). In addition, Wassarman and colleagues published paper in 2015 where they showed that TBI results in blood-brain barrier permeability defects (Katzenberger 2015). Both of these two important papers are not mentioned at all - despite the fact that 130 references are presented. The finding that genes of innate immunity response are upregulated has been made before. The authors should compare their sequence data with the previously published datasets. The specificity of the TRAP-seq method should be documented (e.g. expression of alrm, wunen-2, repo, gliotactin and moody should be compared to neuronal gene expression e.g. nsyb or elav). It should be stated that gliotactin is not a marker for peripheral glial cells but is expressed by the subperineurial glia - as moody and wunen-2 is not a marker for astrocytes but shows a rather broad expression in the adult brain (see Stein Aerts data single cell seq dataset). The finding that CG40470 is the only gene that is persistently downregulated in all days tested is interesting but unfortunately not analyzed in further detail.

Rev. 2:

An extensive literature links TBI to defects in behavior, neural function, and health. However, TBI-induced cellular responses and functional consequences remain an area of intense interest. To this end, van Alphen et al. have developed an effective TBI assay in Drosophila which recapitulates many canonical consequences of TBI, including neuronal death and locomotor impairment. Promisingly, this TBI model is sufficiently robust to elicit significant changes in gene expression in glia, thereby allowing the authors to identify genetic candidates for manipulation. Indeed, disrupting different classes of anti-microbial peptides (AMPs) either improved or impaired resilience to TBI, potentially reconciling previous conflicting studies. Broadly, the authors have designed and validated a novel, precise assay that opens the possibility of screening for TBI response mechanisms and to understand TBI recovery.

Major comments:

1. One complicated feature of TBI is the heterogeneity of outcomes in injured animals, which is also demonstrated in this study. Here, it appears that a large proportion of flies die within ~1-2 days post-TBI, after which it seems that the remaining flies have only a slightly accelerated aging curve. It is appropriate, then, to separate these flies into different groups for analysis. For example, do flies that die rapidly exhibit stronger sleep impairments? If these rapidly-dying flies are separated from the analysis, does the survival curve of the remaining TBI-treated flies look more comparable to sham-treated controls?

2. Figure 2C shows a significant elevation in TUNEL staining to label apoptotic cells in the brain after TBI. Because the authors focus their sequencing studies on glia, it would be informative to test whether the TUNEL-positive cells are neuronal or glial. Do glia activate AMP expression in response to neuronal death? Or to their own apoptosis?

3. The authors have demonstrated in Figure 6 that the mortality and behavioral phenotypes can be partially dissociated. However, this is not consistently addressed. The AMP mutants used in Figure 7 are only tested for mortality; it remains unclear if these AMP mutants, either individually or in combination, could also account for the behavioral phenotypes. Do flies in the "ABC" class from Figure 7 show changes in climbing and sleep behavior after TBI?

4. The authors use TUNEL staining for validation of the TBI model but do not address whether the immune genes that affect survival after TBI also affect neuronal death. To provide a more complete characterization of the role of immune genes, the authors could compare neuronal death in relish mutants and wild-type flies after TBI.

5. The authors claim that the immune response mediates survival after TBI. However, it is possible that immune responses are not playing an active role after TBI but rather that the mutants are more sensitized/susceptible to TBI. The authors should either discuss this caveat in the text or address this experimentally using inducible genetic systems to knock down relish starting ~24h before TBI.

6. Sleep architecture data (Figs. 3D-F) should show daytime and night time results separately.

Minor comments:

1. Are flies that die over the course of the 7-day sleep experiment excluded from the entire dataset, or only from the days after they died? These flies might provide insight into whether the severity of behavioral changes might correlate with mortality.

2. The authors argue that TBI induces an immune response. However, it has also been demonstrated that sleep disturbances themselves can induce the upregulation of immune genes (eg - Dissel et al. 2015; Williams et al. 2007). Therefore, it is possible that TBI might indirectly increase immune responses. by disturbing sleep. This may merit discussion in the text.

3. Figure 7 lacks n values

Dissel S, Seugnet L, Thimgan MS, Silverman N, Angadi V, et al. 2015. Differential activation of immune factors in neurons and glia contribute to individual differences in resilience/vulnerability to sleep disruption. Brain Behav Immun. 47:75-85

Williams JA, Sathyanarayanan S, Hendricks JC, Sehgal A. 2007. Interaction between sleep and the immune response in Drosophila: a role for the NFkappaB relish. Sleep. 30(4):389-400

Rev. 3:

In this manuscript, Van Alphen et al describe a novel model for Traumatic Brain Injury (TBI) and characterize the molecular signatures associated with the injury. While flies have been used as a model to study TBI for some time, the approach described here represents a significant advance over current methodologies because the force and location of the injury can be precisely controlled in an awake animal. The symptoms described, including loss of coordination, shortened life span, and disrupted sleep phenoocpy those in mammals. The manuscript applies cell-type specific RNAseq/TRAPseq to define the molecular signatures of TBI within glial cells and validate a number of genes involved in immune response as contributing to the effects on mortality or function. Overall the manuscript is well written and technically sound. This manuscript will be of broad interest to researchers interested in neurodegeneration, sleep and fly genetics and provides a solid foundation for future studies on TBI. In my opinion, the impact of the manuscript in its current form is sufficient to justify publication. However, the experimental characterization of the assay, and some of the functional validation of TBI-regulated genes could be strengthened. I have included suggestions below, though some may be beyond the scope of this manuscript.

Major comments:

1) Given the novelty of this assay the authors might consider additional descriptions of the behaviors themselves. For example, quantifying differences in behavior immediately following the TBI events through the first few hours of recovery.

2) It is interesting that the behavioral deficits return to normal after a few days. I would be very interested to know if they have memory deficits that persist beyond this point. This may be beyond the scope of the paper, but would be worth discussing.

3) I understand the advantages of targeting the head, even with this precision is it possible to differentiate between neural injury and general stress? Is it possible that targeting a different body region would also lead to climbing/sleep deficits?

4) Genetic background is certainly an important factor for sleep and longevity/aging and is therefore likely very important TBI response. Please describe efforts to account for genetic background.

5) Localizing genes to subpopulations of glia would increase the impact of the findings. would be very helpful to sort TRAP-seq data based on the glial subtype that they express in. I understand this is not entirely straight forward and these data sets don't exist for all glia, but they do for Repo and Alrm, and this alone might be useful. An alternative would be to knock genes down in subsets of glia.

Minor Comments

1) Line 40: Is TBI really one of the leading causes of death? This seems unlikely. Also, (though perhaps too detailed to address here) I imagine most TBI deaths are in elderly patients, which leads me to wonder if the effects of TBI would differ in aged flies.

2) While not critical to the scientific content, Figure 1 could be improved to depict the assay. For example, a cartoon diagraming the components of the assay would be more useful to the image in A, which could be placed in the supplemental figures.

3) Line 258 describes the immediate response of flies to the TBI, and their recovery. Supplemental videos are provided but it would be very useful to quantify this given the novelty of the assay. In addition, the term 'dazed' may inadvertently imply changes in cognitive perception.

4) Figure 3. While not essential, it would be a useful control to show climbing and sleep data in animals given a single strike. Five strikes results in some death, and therefore phenotypes may derive from generalized deficiencies (although the finding that sleep returns to normal after 7 days suggests the effects are specific).

5) Figure 2. When do flies die within the 24hrs following TBI? Is it immediate, or hours after? There are also some caveats about using negative geotaxis to infer sensory-motor function (though these are likely shared in rotarod studies). It does not rule out things like general arousal, endurance, or motivation.

6) Recent work from the Donlea group showed that antennal axotomy results in increased sleep. It is worth commenting on the difference in sleep phenotypes that result from each type of neural injury.

7) How was the strength of the TBI-inducing stimulus chosen?

8) In some cases the language could be more precise. E.g. line 422 'Also, quite a few members of the turanadot…'

Rev. 4:

In this manuscript ("Glial immune-related pathways as mediators of closed head TBI effects on behavior and lethality in Drosophila"), van Alphen and Stewart et al. describe an in vivo traumatic brain injury (TBI) model in adult fruit flies that triggers a number of quantifiable pathological and behavioral phenotypes associated with nervous system decline. The authors have performed a large scale unbiased transcriptional screen, utilizing Translating Ribosome Affinity Purification and Sequencing (TRAP-Seq) to identify genes upregulated in glial cells at several time points post-TBI. In follow-up validation experiments, they show that specific genes (NF-kB transcription factor relish and classes of secreted AMP immune factors) differentially influence TBI-induced phenotypes. Overall, this work offers an interesting new model and, most importantly, a novel data set (TRAP screen) for the glial/injury scientific community to explore and potentially define additional glial-specific signaling pathways that are invoked after closed head injury (both beneficial and detrimental pathways). However, there are some questions and issues related to analysis and methods that must be addressed before the article would be appropriate for publication.

Specific Comments for Authors:

1. The primary strength of this manuscript lies in the TRAP screen. Thus, the authors must provide a complete/comprehensive data set of the genes altered in the TBI assay. In addition, this list of differentially expressed genes with altered expression levels, p values, etc. will provide a complete picture of the efficacy of the TRAP approach. This manuscript does show that known adult glial-specific genes are upregulated in TBI animals, but, as presented, it is not clear if neuron-specific genes are also induced (and/or detected in uninjured animals due to technical/specificity challenges).

2. The authors state that differential gene expression analysis was identified using a p value of 0.1 (and Log2 value of 0.6) as a threshold. This value is high. The authors should clarify why this is an appropriate cutoff value (or choose to make the cutoff more stringent, for example p value of 0.05 or below).

3. It is still unclear to this reviewer how significance of 7 day RNAseq data samples were generated with an n value of 2. There are trend analysis statistical approaches that can be utilized (which would, for example, track increased gene expression at days 1, 3, and 7 after injury), but it's not clear in the Methods if this type of approach was utilized.

4. Fast QC data from Illumini Seq runs should be included in the Results.

5. The authors should include a discussion of the recent publication (Saikumar et al., PNAS, July 2020), which describes a similar TBI protocol and neurodegeneration analysis in adult Drosophila.

6. In Figure 2D, are these TUNEL-positive values in the entire central brain - or optical sections?

7. This manuscript aims to explore the role of glia in TBI-induced changes in physiology. This model will be substantially strengthened if the authors can perform a subset of experiments that specifically alter the expression of genes identified in the TRAP screen (for example, components of the NF-kB pathway) and show they (at least partially) recapitulate the phenotypes of whole animal mutants. It is not surprising that this would be difficult to perform with the AMP factors due to functional redundancy.

8. The authors provide a range of behavioral readouts for TBI in control versus NF-kB mutants. It would be informative to complement these results with a physiological analysis. For example, how is cell death (TUNEL) affected? Are there changes in synaptic connectivity (e.g. nc82 staining), which often precede cell death in neurodegeneration models.

9. Minor point: Authors should describe head homogenization, step pre-bead incubation, and RT details (cDNA synthesis) in greater detail.

---

## [Decision Letter · Decision Letter 2]

25 Mar 2021

Dear Dr Allada,

Thank you for submitting your revised Research Article entitled "Glial immune-related pathways as mediators of closed head TBI effects on behavior and lethality in Drosophila" for publication in PLOS Biology. Thank you also for your patience as we completed our editorial process, and please accept my apologies for the delay in providing you with our decision. I have now obtained advice from two of the original reviewers and have discussed their comments with the team of editors. 

The reviews are attached below. You will see that Reviewer 1 remains negative, however we do think you have discussed satisfactorily the issues, as we requested. Based on the reviews, we will probably accept this manuscript for publication, provided you address the data and other policy-related requests included at the end of the letter. In addition, we would like to make some suggestions to improve the title:

Glial immune-related pathways mediate effects of closed head traumatic brain injury on behavior and lethality in Drosophila.

We expect to receive your revised manuscript within two weeks. 

-  a cover letter that should detail your responses to any editorial requests and whether changes have been made to the reference list

*Published Peer Review History*

Please note that we would like you to make the peer review history publicly available. The record will include editor decision letters (with reviews) and your responses to reviewer comments. Please see here for more details:

*Early Version*

Sincerely,

The *PLOS Biology* Editors

Fig. 2A, B, D; Fig. 3A-F; Fig. 4A, B, D; Fig. 5; Fig. 6A-F; Fig. 7A-G; Fig. S1B-F; Fig. S2A-I; Fig. S3A-D; Fig. S4A-D; Fig. S5A-F; Fig. S6; Fig. S7A, B; Fig. S8A-M; Fig. S9A-C

***Please ensure that your Data Statement in the submission system accurately describes where your data can be found and also that mentions that Fast QC data has been uploaded to the GEO repository (Series record GSE164377). Please note that this data should be made publicly available at this time, before the manuscript enters production.

DATA NOT SHOWN

Please note that per journal policy, we do not allow the mention of "data not shown", "personal communication", "manuscript in preparation" or other references to data that is not publicly available or contained within this manuscript. Please either remove mention of these data or provide figures presenting the results and the data underlying the figure(s).

Reviewers' comments:

Rev. 1:

I found the rebuttal letter as well as the revised version of the manuscript disappointing. As stated before, Gliotactin is a protein found on all subperineurial glial cells where it is involved in septate junction formation. The essence of the paper is to introduce a new method to induce TBI. Unfortunately, possible other tissue damage is not analyzed. The study then conducts a RNAseq analysis and further analyses the role of NF-Kb in TBI. The results (increased mortality) are not easy understandable and lead to the conclusion that innate immune responses to TBI can have beneficial and detrimental effects, which again is not further analyzed. Given the lack of novelty, clarity and the lack of further information on the novel gene function CG40470 in response to TBI I wonder why this manuscript should be published in a high-profile journal.

Rev. 4:

The authors have addressed my major concerns with the manuscript. The authors acknowledge that the use of two replicates for one of the timepoints is not ideal, but they are transparent about this result and I don't feel it should prevent publication of the manuscript.

---

## [Editor Report · Decision Letter 3]

30 Sep 2021

Dear Dr Allada,

Thank you for submitting your revised Research Article entitled "Glial immune-related pathways mediate effects of closed head traumatic brain injury on behavior and lethality in Drosophila" for publication in PLOS Biology. Thank you also for your patience while we were checking the revision.

We have now reviewed the new version of the manuscript in light of all the updates you have made and, while we will probably accept the manuscript for publication, we need you to provide further clarifications for some of the recent changes and to correct some errors in the figures and the figure legends:

- Fig. 4A: While in the text and the figure legend of this figure indicate that glia-specific genes are enriched and that neuron-specific genes elav and nsyb are not significantly changed, in the figure the neuronal genes have been removed and two new glia-specific genes, moody and wun2, have been added. Please explain why these changes have been done and update accordingly the figure legend and the figure if necessary, or correct the mistakes.

- Fig. 4B: While the figure legend has not changed, the days after TB1 treatment are now 1, 2 and 3, instead of 1, 3 and 7. Please update the figure if this is a mistake.

- Fig. 4C: In the cover letter of the revision, you mention that there was a minor error in the Venn diagram and that the figure had been updated, however the number of genes differentially expressed in Day 1, 3 and 7 have changed substantially and are now much lower. Please provide a suitable explanation for these changes.

- Fig. S4: All the graphs showing that sleep is largely unaffected in flies dying within 7 days post-TBI treatment have changed quite substantially from the previous version and you do mention in the cover letter that the flies dying do not exhibit strong sleep effects after TBI compared to their surviving counterparts. Please explain the reason why the results and graphs have changed. The number of flies reflected in the experiments has significantly changed: n=17 sham-treated flies (instead of 84) and n=40 TBI flies (instead of 61). Please explain why the number of flies is now much lower.

- Fig. S9: Thank you for repeating the experiment and providing the underlying data for sections A and B. We noticed that you have changed the title in the figure and the conclusion in the text from “TBI does not affect climbing in ΔAMP null mutants, but increases sleep” to “ΔAMP null mutants exhibit decreased climbing and increased sleep after TBI.” As the data supports this conclusion, we assume this was a mistake in the previous version – please explain.

- In the introduction, the previous version mentioned that “CRISPR deletions of most AMP classes increase TBI-induced mortality, but survival is increased in flies lacking both Defensin and the four classes of Cecropins, suggesting that the innate immune response to TBI in Drosophila can have both beneficial and detrimental effects.” In addition, in the results stated that “Surprisingly, flies only lacking class A show increased survival (Fig. 7E, A, p = 0.026), suggesting that Cecropins and/or Defensin upregulation is detrimental rather than beneficial for TBI survival.” However, the first sentence has been removed in the new version and in the second one, only Defensin is mentioned. Please explain these changes.

We expect to receive your revised manuscript within two weeks. 

Sincerely,

The *PLOS Biology* Editors

---

## [Editor Report · Decision Letter 4]

22 Oct 2021

Dear Dr Allada,

On behalf of my colleagues I am pleased to say that we can in principle offer to publish your Research Article entitled "Glial immune-related pathways mediate effects of closed head traumatic brain injury on behavior and lethality in Drosophila" in PLOS Biology, provided you address any remaining formatting and reporting issues. These will be detailed in an email that will follow this letter and that you will usually receive within 2-3 business days, during which time no action is required from you. Please note that we will not be able to formally accept your manuscript and schedule it for publication until you have made the required changes.

Sincerely, 

The *PLOS Biology* Editors